# Spatially resolved transcriptomic profiling of degraded and challenging fresh frozen samples

Reza Mirzazadeh[1,9], Zaneta Andrusivova[1,9], Ludvig Larsson [1,9],
Phillip T. Newton [2], Leire Alonso Galicia[1], Xesús M. Abalo [1], Mahtab Avijgan[2],
Linda Kvastad [1], Alexandre Denadai-Souza [3], Nathalie Stakenborg [3],
Alexandra B. Firsova[4], Alia Shamikh[5,6], Aleksandra Jurek[7], Niklas Schultz[8],
Monica Nistér [8], Christos Samakovlis[4], Guy Boeckxstaens [3] &
Joakim Lundeberg [1] ✉

Spatially resolved transcriptomics has enabled precise genome-wide mRNA expression profiling within tissue sections. The performance of methods targeting the polyA tails of mRNA relies on the availability of specimens with high RNA quality. Moreover, the high cost of currently available spatial resolved transcriptomics assays requires a careful sample screening process to increase the chance of obtaining high-quality data. Indeed, the upfront analysis of RNA quality can show considerable variability due to sample handling, storage, and/or intrinsic factors. We present RNA-Rescue Spatial Transcriptomics (RRST), a workflow designed to improve mRNA recovery from fresh frozen specimens with moderate to low RNA quality. First, we provide a benchmark of RRST against the standard Visium spatial gene expression protocol on high RNA quality samples represented by mouse brain and prostate cancer samples. Then, we test the RRST protocol on tissue sections collected from five challenging tissue types, including human lung, colon, small intestine, pediatric brain tumor, and mouse bone/cartilage. In total, we analyze 52 tissue sections and demonstrate that RRST is a versatile, powerful, and reproducible protocol for fresh frozen specimens of different qualities and origins.

Spatially resolved transcriptomics (SRT) is a set of technologies used to chart genome-wide mRNA expression within tissue sections, and it has become widely used in genomics research in the past decade[1–3]. SRT has opened up new possibilities to explore the spatial architecture of cells and their interactions in the tissue context,

exemplified by works in neuroscience[4], developmental biology[5], and disease[6,7].

The first report of next generation sequencing (NGS) based SRT method for high throughput spatial mRNA profiling was published in 2016[8]. This work paved the way for unbiased capturing of whole

---

[1]Department of Gene Technology, KTH Royal Institute of Technology, Science for Life Laboratory, Stockholm, Sweden. [2]Department of Women's and Children's Health, Karolinska Institute, Solna, Sweden and Astrid Lindgren Children's Hospital, Karolinska University Hospital, Solna, Sweden. [3]Department of Chronic Diseases and Metabolism, Katholieke Universiteit te Leuven, Leuven, Belgium. [4]Department of Molecular Biosciences, Wenner-Gren Institute, Stockholm University, Science for Life Laboratory, Stockholm, Sweden. [5]Department of Oncology-Pathology, Karolinska Institutet, Stockholm, Sweden. [6]Department of Clinical Pathology and Cytology, Karolinska University Hospital, Stockholm, Sweden. [7]10x Genomics, Stockholm, Sweden. [8]Department of Oncology-Pathology, Karolinska Institutet, BioClinicum, Karolinska University Hospital, Stockholm, Sweden. [9]These authors contributed equally: Reza Mirzazadeh, Zaneta Andrusivova, Ludvig Larsson. ✉e-mail: joakim.lundeberg@scilifelab.se

transcriptomes from tissue sections. The underlying principle of this technology is a dense grid of spatially barcoded oligo(dT) probes printed on a microscope glass slide, which can be used to capture the polyA tails of mRNA molecules from a tissue section, thus facilitating spatially resolved gene expression profiling. The tissue section is also stained and imaged with a microscope, which makes it possible to combine gene expression profiling with histology. Currently, the most broadly used NGS with spatial barcoding platform is Visium (10× Genomics)[3], an updated version of the same principles presented by Ståhl et al.,[8] currently with 5000 barcoded spots, each with a diameter of 55 μm (see 10× Genomics webpage https://www.10xgenomics.com/).

In this work, we use the Visium protocol, which is currently optimized for fresh frozen (FF) tissue specimens and recommends a RIN (RNA Integrity Number) score higher than or equal to 7. RIN is a critical metric to assess the quality and level of RNA degradation before starting an SRT experiment[9,10]. FF samples are the preferred choice for unbiased polyA-based SRT technologies due to their high preservation of polyadenylated transcripts. However, a major limitation with polyA-based SRT is its reduced ability to process degraded samples. In spite of the widespread use of Visium for FF samples, there is a need for a method that works well on samples with low RNA quality. Recently, 10× Genomics introduced a new chemistry for Formalin-Fixed Paraffin-Embedded (FFPE) samples. In FFPE samples, it is well documented that RNA molecules are fragmented, where the degradation often affects the polyA tails of the RNA[11,12]. To overcome the aforementioned issue, the FFPE SRT approach relies on a gene-panel to target and capture protein-coding regions of the transcriptome instead of targeting the polyA tails.

Based on this recent development, we propose a strategy for spatial analysis of FF tissue specimens with moderate/low RIN scores, that we name RNA-Rescue Spatial Transcriptomics (RRST). This protocol makes use of the same targeted gene-panel that was designed for FFPE material with additional modifications to work on FF tissues, including a gentle formalin fixation step and a baking step to improve tissue adherence to the slide surface. We demonstrate the capabilities of our RRST method by profiling the tissue transcriptomes of a variety of biological specimens, and comparing the results with data generated by the standard Visium protocol.

## Results

### RRST implementation in fresh frozen tissue sections

We attempted to make the Visium SRT technology compatible for analysis of degraded FF samples by introducing specific modifications to the commercially available Visium FFPE protocol. In the original FFPE protocol, tissue sections are first deparaffinized through a series of washes with xylene/ethanol. Then, the tissue sections are stained with hematoxylin-eosin and de-crosslinked in the Tris-EDTA (TE) buffer at 70 °C for an hour. The sections are then incubated with probe sets that hybridize in pairs to each transcript, targeting approximately 19 K protein-coding genes. Upon correct probe hybridization to mRNA transcripts, each pair is ligated to one another and captured by oligo(dT) probes attached to the surface of the glass slide, where spatial barcodes are introduced through a cDNA synthesis step. The cDNA molecules, which now hold information about the target transcript and its spatial location, are released from the slide surface for final library preparation and sequencing.

It should be noted that deparaffinization of FFPE samples followed by staining procedure increases the chance of tissue detachment due to the repeated washing steps in the initial steps of section processing, which might result in low quality/failed data generation (Supplementary Fig. 1 and Supplementary Video 1). Furthermore, FFPE specimens are usually heavily crosslinked due to a prolonged formalin fixation process, and thus crosslink reversal is a critical step to access the RNA molecules within tissue sections. This reversal is done through long incubation at high pH and temperature. We speculated that the

long decrosslinking incubation step used in FFPE protocol may potentially lead to RNA degradation in the shortly-fixed FF samples.

In the RRST protocol, FF tissue sections are fixed with formalin, instead of methanol, for 10 min at room temperature, followed by a baking step of 20 min at 37 °C, which we found necessary in order to improve tissue section adhesion to the Visium slides. The cross-linking reversal step is removed to prevent RNA degradation, which in addition shortens the duration of the protocol by an hour. A detailed workflow of the RRST is depicted in Supplementary Fig. 2 and a step-by-step protocol can be found in the Methods section.

### Performance of RRST in high quality FF samples

We first set out a test to evaluate how well the RRST protocol performs on two FF samples with high RIN values: a mouse brain sample (RIN 8.8) and a human prostate tumor specimen (RIN 10) (Supplementary Table 1). The mouse brain has become the sample of choice to benchmark SRT technologies because of its well-defined anatomical structures, which have been characterized in detail based on histology and spatial gene expression[13]. We performed both RRST and standard Visium on sections collected from the same tissue blocks for both mouse brain and prostate tumor samples. The sequencing saturation was comparable between the two protocols (Supplementary Data 1). We found that RRST can profile the tissue transcriptome with approximately two-fold increase in the number of detected genes per spot compared to the standard protocol (Fig. 1b, c). Moreover, in both the mouse brain and prostate tumor samples, we observed a high concordance (Pearson $R = 0.82$, $p < 2.2e{-}16$ and $R = 0.76$, $p < 2.2e{-}16$) between the aggregated gene counts across the two datasets, excluding genes that were not targeted by the RRST panel (Fig. 1d). This indicates that the data obtained with the RRST approach display a high similarity with the data obtained with the standard Visium protocol. However, the probe panel used for RRST excludes certain transcripts, such as those transcribed from mitochondrial genes, ribosomal protein coding genes or ncRNAs (Supplementary Fig. 3). With the exception of these three RNA types, the majority of detected transcripts come from protein coding genes and are detected with both methods (Supplementary Fig. 3), although at drastically different UMI counts and detection rates (Fig. 1d). For the majority of transcripts, RRST protocol appears to exhibit a higher capture efficiency.

### RRST recovers spatial transcriptomics data from challenging FF samples

There is a growing number of studies using the standard Visium platform for FF tissues to address biological questions[14]. The assessment of RNA quality through RIN measurement (RIN ≥ 7) is suggested as an important criterion to define the quality of tissues for successful spatial gene expression profiling. In our own experience, some tissue types are more challenging to retrieve good/high quality Visium data from. There could be several factors contributing to low/moderate RNA quality, such as intrinsic biological characteristics of the tissue, rapid RNA degradation upon surgical procedure or sensitivity to freezing/thawing during tissue sectioning. Hence, we aimed to apply RRST to some challenging tissue types that are known to perform poorly using the standard 3′ capture Visium platform (Supplementary Figs. 4, 5).

### Adult human lung tissue

To date, spatial transcriptome profiling of human lung tissue has rarely been investigated[15]. Based on our own experience, FF mouse and human lung tissue samples are highly challenging to process with the standard Visium protocol. Therefore, we tested the performance of RRST in FF healthy adult human lung samples (Fig. 2a and Supplementary Fig. 6a) retrieved from two patients (LNG1, RIN 6.8 and LNG2, RIN 7.1, Supplementary Table 1), where the 3′ capture protocol

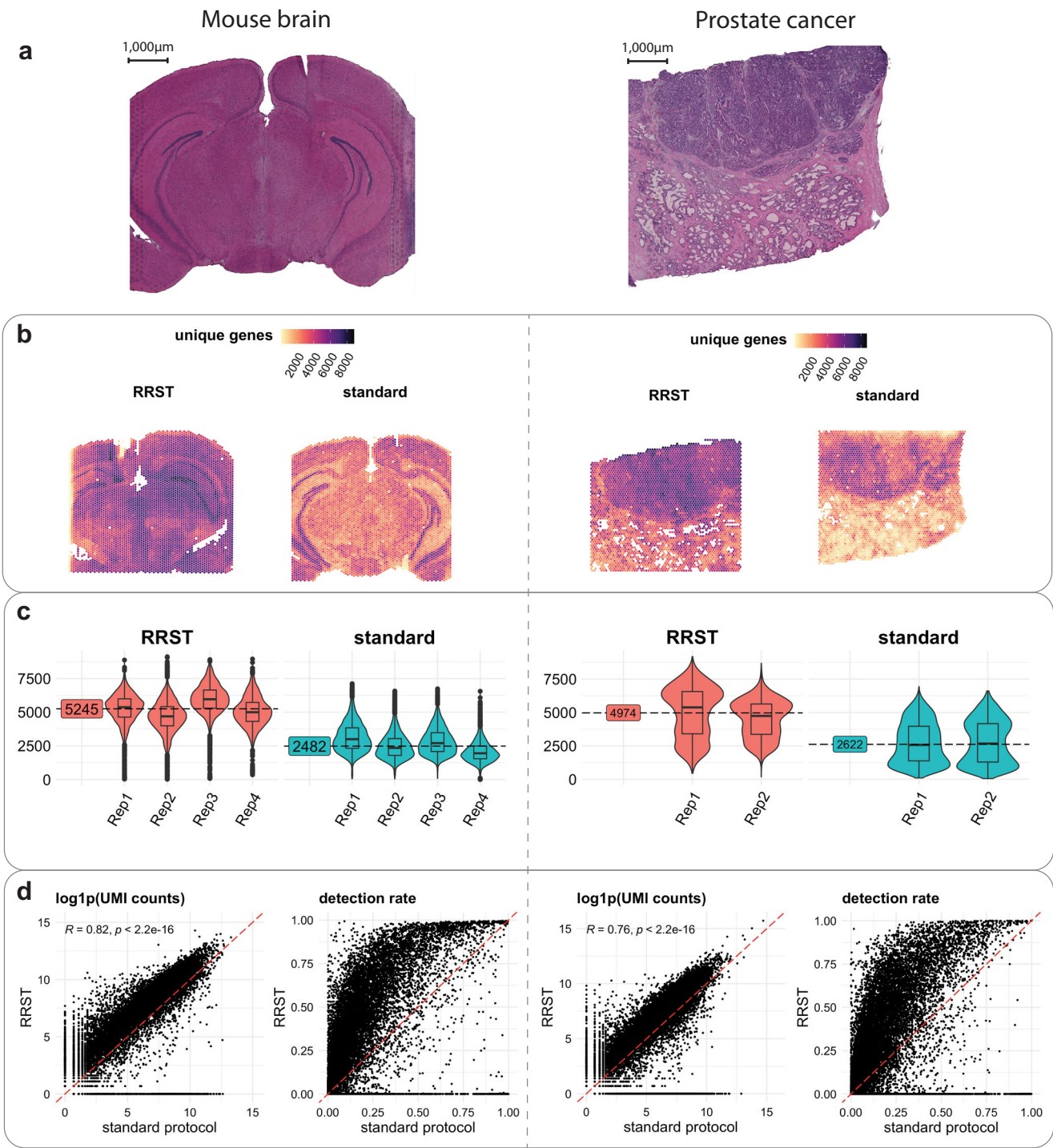

**Fig. 1 | Comparison of RRST and Visium on mouse brain and prostate tumor samples. a** H&E images of a representative tissue section from mouse brain (left) and prostate cancer (right). The entire dataset consisted of 8 consecutive mouse brain tissue sections and 4 consecutive prostate cancer tissue sections. Half of the tissue sections were processed with RRST and the remaining half with standard Visium protocol. **b** Spatial distribution of unique genes in two representative tissue sections for each tissue type, one processed with the RRST protocol and one processed with the standard Visium protocol. **c** Distributions of unique genes per spot visualized as violin/box plots colored by experimental protocol for mouse brain and prostate cancer data. Box plots are presented as median values where the lower and upper bounds are the 25th and 75th percentiles. The upper and lower limits of the boxplots are defined by the closest value no further than 1.5*IQR

(inter-quartile range) from the closest bound. Values outside of the upper and lower limits are highlighted as outliers. The median number of unique genes is highlighted for each group (sample type and protocol) next to the violin plots. **d** gene-gene scatter plots between RRST data (*y*-axis) and standard Visium data (*x*-axis) of log1p-transformed UMI counts and detection rates using the data shown in (**b**). The UMI counts and detection rates were calculated across the pooled technical replicates within each experimental protocol. The red dashed line highlights a 1-to-1 relationship. For the log1p-transformed UMI counts scatter plot, only genes targeted by the probe panel were included. The detection rate for a gene is defined as the proportion of spots with detected UMI counts. The statistical test is based on the Pearson product moment correlation coefficient and *p*-values were estimated using a two-sided alternative hypothesis.

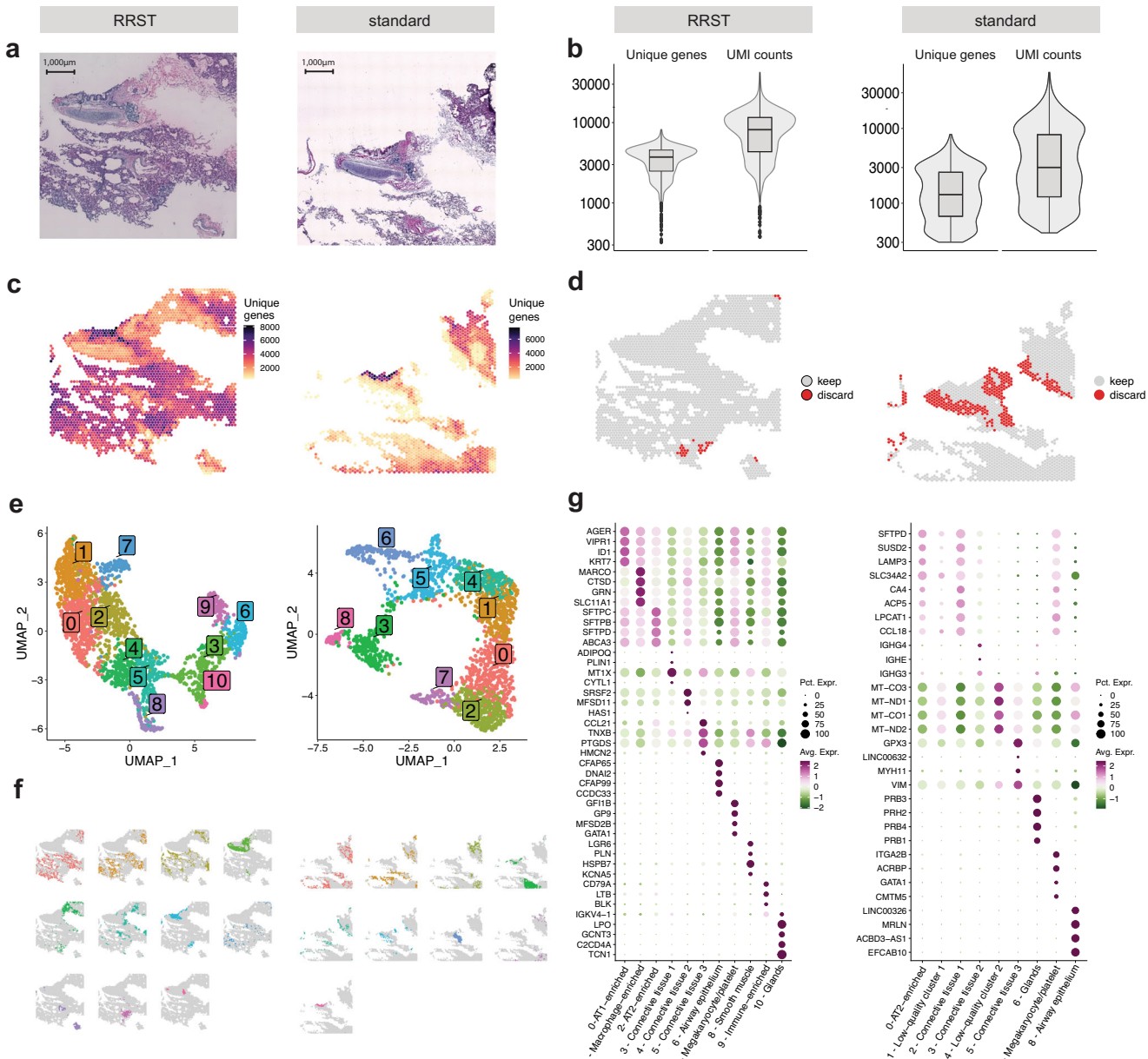

**Fig. 2 | RRST and standard Visium applied to human adult lung tissue.** Each subplot shows the RRST data on the left side and the standard Visium data on the right side. **a** H&E images of two representative tissue sections collected from the same tissue block. **b** Violin/box plots showing the distribution of unique genes and UMI counts for RRST (*n* = 1) and standard Visium (*n* = 2) data generated from consecutive tissue sections from the same lung tissue specimen. The *y*-axis is shown in log10 scale. Box plots are presented as median values where the lower and upper bounds are the 25th and 75th percentiles. The upper and lower limits of the box-plots are defined by the closest value no further than 1.5*IQR (inter-quartile range) from the closest bound. Values outside of the upper and lower limits are high-lighted as outliers. **c** Unique genes per spot mapped on tissue coordinates. **d** Spatial visualization showing what spots were discarded due to low quality (less than 300 unique genes detected). **e** UMAP embedding of adult lung data colored by clusters detected by unsupervised graph-based clustering (louvain). **f** Split view of clusters (same as in **e**) mapped on tissue coordinates. **g** Dot plots of the top marker genes for each cluster. Each cluster was annotated based on its spatial localization in the tissue and expression of canonical marker genes.

performed poorly. Our RRST method detected roughly a 2-fold and 10-fold increase in the number of detected genes per spot in these two patients respectively, indicating the robustness and power of RRST to profile gene expression spatially in challenging tissue types (Fig. 2b, c and Supplementary Fig. 6b, c). As a quality control step, spots with few unique genes detected are commonly discarded based on an empirical cutoff threshold, where thresholds between 500 and 1000 unique genes are common. Here we used a softer cutoff threshold of 300 unique genes to include as many spots as possible from both conditions (Fig. 2d and Supplementary Fig. 6d). Even with this soft cutoff threshold, ~21–80% of the spots were discarded for the standard

Visium data, while only 1.3–2.3% of spots for the RRST data from the same tissue blocks (Supplementary Fig. 7). For downstream analysis, we first focused on one of the patients (LNG1). A common practice in exploratory SRT data analysis is to perform data-driven clustering followed by marker detection using differential expression analysis (DEA). Clusters identified from SRT data typically represent groups of spots that share similar cell type composition. We reasoned that by performing this type of exploratory analysis on the two data types, with the same parameter settings, we can get an idea about how the difference in data quality affects interpretation. After dimensionality reduction and clustering, we detected 11 clusters in the RRST and 9

clusters in the standard Visium data (Fig. 2e, f). Notably, marker detection by DEA highlighted distinct marker genes for each of the 11 RRST lung clusters, whereas clusters 0, 1, and 2 in the standard Visium lung data were difficult to distinguish from each other (Fig. 2g). Moreover, cluster 4 in the standard Visium lung data displayed differential expression of mitochondrial transcripts, which is indicative of low quality transcriptomic profiles[16]. Some of the top markers detected for cluster 8 (airway epithelium) in the standard Visium lung data were ncRNAs (*LINC00326*, *ACBD3-AS1*, and *AC023300.2*), which RRST does not detect, and therefore the use of this targeted approach can be a limiting factor for certain types of analysis. We characterized the LNG1 clusters (RRST and standard Visium) based on the top markers and their spatial localization (Fig. 2f, g). Next, we inspected the expression of top marker genes detected across both conditions in the following four selected clusters: airway epithelium, megakaryocyte/platelet-enriched, smooth muscle, and glands. The number of unique genes detected were higher for RRST in gland, megakaryocyte/platelet-enriched, and smooth muscle clusters, but not in the airway epithelium cluster (Supplementary Fig. 8a). We found that the shared marker genes were more consistently expressed in the RRST data, except for the airway epithelium which displayed comparable expression levels across the two conditions (Supplementary Fig. 8b). In addition, the detection rates of DEGs within each cluster displayed a similar trend (Supplementary Fig. 8c), reiterating that the increase in data gained with RRST strengthens the signal of region-specific marker genes. We also applied the same analysis workflow on the second patient sample (LNG2), and observed similar trends with clearer biological signal in RRST data compared to standard Visum data (Supplementary Fig. 6). Although the results of data-driven clustering are influenced by a number of different parameters (not only data quality) this comparison suggests that the higher quality RRST data makes it easier to characterize molecular profiles of lung tissue sections using popular exploratory analysis methods.

## Adult human colon tissue

Next, we investigated if the RRST protocol can be used to obtain spatial gene expression data from samples for which the standard Visium failed. For this purpose, we investigated FF adult human colon samples collected for the Gut Cell Atlas consortium. After extensive efforts to generate good-quality data from these tissue blocks, we could conclude that the intestinal epithelial tissues are particularly susceptible to mRNA degradation and consequently difficult to process with the standard Visium protocol. It is of note that the gut is a highly delicate tissue that is filled with digestive enzymes and a microbiome of varying quality and quantity, which in turn can lead to a rapid degradation of RNA[17]. We processed colon tissue sections obtained from two patients (Fig. 3a) with moderate RNA integrity (RIN of 4.5 and 5.1, Supplementary Table 1). To assess whether mRNA degradation differs between tissue types, we manually annotated the data into three major regions: mucosa, submucosa, and muscularis (Fig. 3a). Notably, in the standard Visium data, we observed low numbers of unique genes and UMI counts in the cell dense epithelial layer (mucosa), while we could still recover decent numbers of unique genes and UMIs counts in the muscularis (Fig. 3b–d). This observation was in line with what has been reported previously in literature, where it was shown that mRNA degrades more rapidly in the intestinal epithelium compared to the intestinal muscle tissue[17]. However, with the RRST protocol, we were able to recover good-quality data both from the mucosa and sub-mucosa in tissue sections collected from the same OCT block (Fig. 3b–d). The RRST method generated more even data coverage across different tissue regions (Fig. 3b), indicating that the method is able to mitigate the effects of tissue-specific degradation. To demonstrate the effect of tissue-specific degradation, we investigated expression in the mucosa of 11 intestinal epithelial markers (Fig. 3e) selected from the Gut Cell Atlas[18]. These results show that the RRST

data provided higher detection rates and more even expression values, thus indicating that the method can be used to profile regions with degraded mRNA.

## Adult human small intestine tissue

The mRNA quality of FF tissue blocks depends on a number of different factors, such as sample collection, handling, and storage[19]. To estimate the overall quality of a specimen, it can be useful to measure RIN and/or DV200. However, for certain sample types, we have observed that mRNA can degrade rapidly even when they are properly stored in freezers, which in turn means that quality measurements become less reliable over time. One such sample, where we could observe a rapid degradation, was a FF OCT-embedded tissue specimen from an adult human small intestine (Ileum) obtained from the Gut Cell Atlas project. Approximately one month after sample collection, we processed four tissue sections from the FF OCT block using the standard Visium protocol, which generated high-quality data from all tissue regions: mucosa, Tertiary Lymphoid Tissue (TLS), submucosa, muscularis and serosa (Fig. 4a, b and Supplementary Fig. 9). Surprisingly, when we repeated the experiment using the same tissue block six months later (eight tissue sections), we observed an almost complete loss of gene expression data in the mucosal/submucosal layers, while the data in the muscularis remained stable and comparable to the first experiment (Fig. 4b). These results reiterate what we observed in the adult human colon tissue, that mRNA degradation can vary in different tissue types and even within the same section, which cannot be assessed by bulk RIN quality check prior to the SRT assay. Moreover, it became clearer that the main challenge with running Visium on intestinal lower GI tract epithelial tissues is the rapid mRNA degradation. To test whether we could use our RRST method to recover high-quality data from the same block, we processed two tissue sections from the same OCT block approximately two years after sample collection (RIN 7.8, Supplementary Table 1). As indicated by the number of unique genes, we were able to detect higher numbers in the mucosa and submucosa compared to the second attempt, albeit with lower numbers than the initial experiment conducted two years earlier (Fig. 4b). Notably, the second attempt with the standard Visium method (~ six months after sample collection) resulted in an average of 159 unique genes per spot in the mucosa, whereas the RRST data (~ two years after sample collection) resulted in an average of 814 unique genes per spot in the mucosa (Fig. 4b). Moreover, a large fraction of the expression data obtained with standard Visium comes from mitochondrial transcripts, ribosomal protein coding transcripts or lncRNA, which are commonly filtered out prior to downstream analysis, whereas RRST only targets protein coding genes (Fig. 4c). Next, we looked closer at the mucosa region to determine how the difference in quality affects the expression profiles. For each time point and gene, we plotted its average expression against its detection rate for spots annotated as mucosa (Fig. 4d.). We found that the initial dataset (~1 month after sample collection) provided both the highest detection rates and expression levels. The detection rates and expression levels dropped substantially in the second dataset (~6 months after sample collection) and were partially recovered in the third dataset generated with RRST after ~2 years. Next, we took five enterocyte markers from the Gut Cell Atlas and visualized their expression across the tissue sections in the three datasets. These markers were clearly visible in the mucosa in the first dataset and the RRST dataset but not in the second dataset (Fig. 4e). Based on these results, we speculate that gut epithelial tissues contain high amounts of RNAses and, therefore, repetitive freeze/thaw cycles and long-term storage lead to mRNA degradation, hence our RRST approach can help overcome these effects. Overall, these results demonstrate that our RRST protocol can be used in FF samples with low/moderate RNA integrity and to recover data from FF tissue blocks that have been stored for long periods of time.

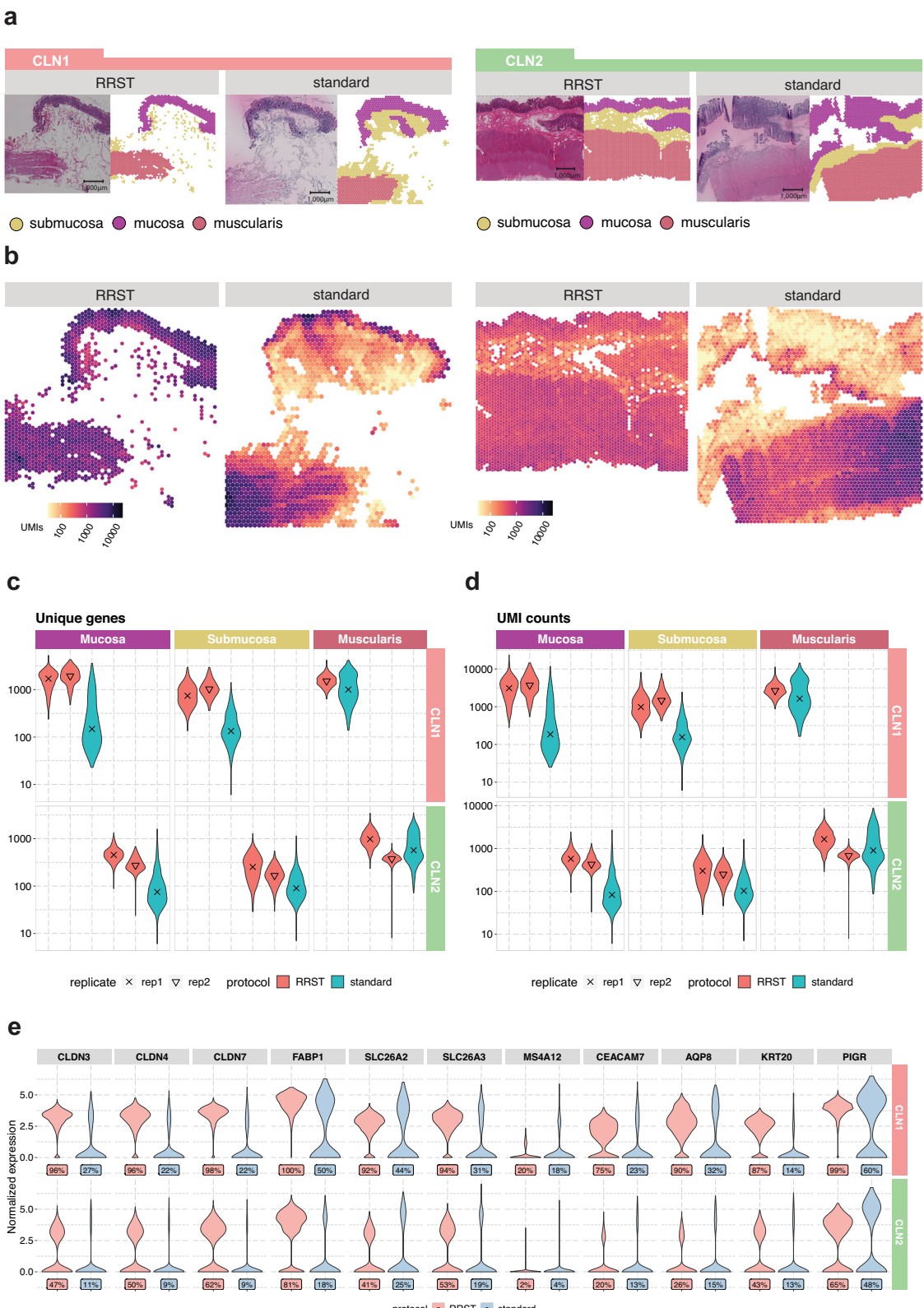

**Fig. 3 | Comparison of data quality in RRST and standard Visium datasets generated from adult human colon tissues. a** Representative H&E images and annotated regions for two patient samples processed by either RRST ($n = 4$) or standard Visium ($n = 2$) protocol. The spots in each tissue section were labeled into three categories: mucosa, submucosa, and muscularis. **b** Distribution of UMI counts in the tissue sections shown in (**a**). The color scale represents log10-transformed counts. **c** Distribution of unique genes per spot in the three annotated regions (mucosa, submucosa, and muscularis) visualized as violin plots, for all

tissue sections. The y-axis shows log10-transformed counts. **d** Distribution of UMI counts per spot in the three annotated regions (mucosa, submucosa, and muscularis) visualized as violin plots. The y-axis shows log10-transformed counts. **e** Expression of 11 epithelial markers in the mucosa for the two adult colon samples visualized as violin plots. A comparison between the two protocols is shown for each gene and the corresponding detection rate is highlighted below each violin plot. The detection rate is defined as the percentage of spots (in the mucosa) where the gene is detected.

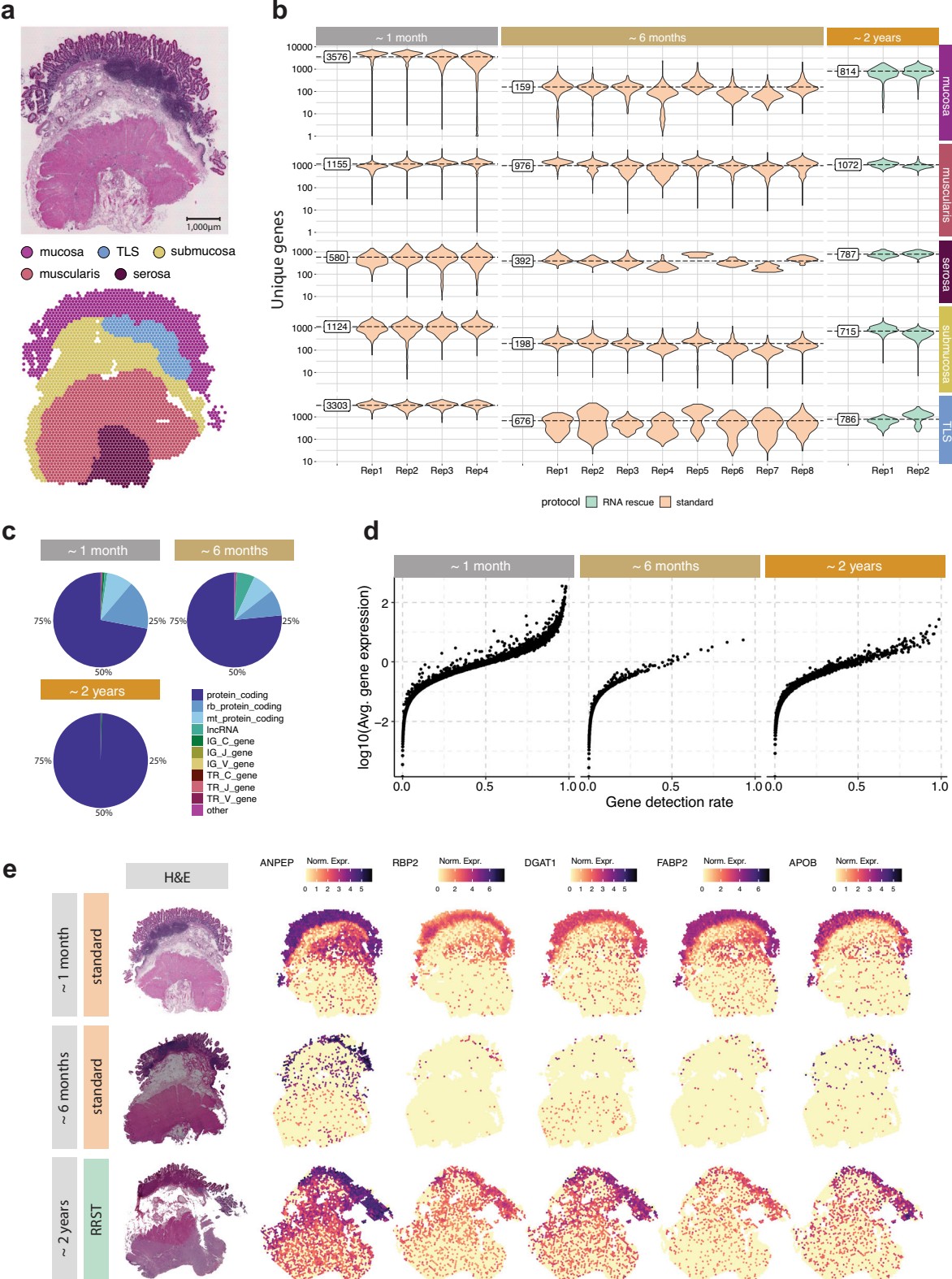

## RRST revives spatial transcriptome profiles of precious clinical samples

Spatial gene-expression profiling of clinical samples can enable discoveries required to develop new strategies for early diagnosis and individualized therapies at molecular levels[20]. Treatment of pediatric brain tumors is continually being improved upon; however, there is a great need for new treatment options. Due to the limited amount of tissue available for research, there is usually not enough material for

tissue optimization and RIN measurement to assess whether the sample quality is sufficient for the standard 3' capture protocol. In order to investigate how RRST performs in such precious clinical samples, we processed two pediatric brain tumor specimens (RIN 7.0 and 7.1, Supplementary Table 1) from which we had previously failed to generate data using the standard 3' polyA capture protocol. In contrast to previous samples described in this study, the pediatric brain tumor samples passed the recommended RNA quality threshold for the

**Fig. 4 | Comparison between RRST and standard Visium on an adult human small intestine sample over time. a** Representative H&E image (top) and spots colored by five major tissue regions (bottom): mucosa, TLS, submucosa, muscularis, and serosa. TLS, Tertiary Lymphoid Tissue. The full small intestine dataset consisted of 14 tissue sections collected from the same specimen at different time points. Only sections collected at the last point were processed with RRST, while the other sections were processed with standard Visium protocol. **b** Overview of data quality in the five annotated tissue regions over time, visualized by violin plots of the number of unique genes per spot. The time points represent the approximate storage time after sample collection: ~1 month, ~6 months, and ~2 years. Replicates obtained for each time point are shown on the *x*-axis. The fill color of the violin plots indicates the applied protocol. For each time point, labels on the left

side of the violin plots represent the average over all replicates. **c** RNA biotype content for the three datasets visualized as a pie chart. Proportions represent the UMI counts detected for each biotype. The targeted RRST data include protein coding, immunoglobulin, and T-cell receptor transcripts. **d** Mean-detection rate relationship in the mucosa for data collected at the three different time points. The *y*-axis shows log10-transformed average number of UMIs for each gene, and the *x*-axis shows the detection rate for each gene. The detection rate is defined as the fraction of spots where the gene is detected. **e** Spatial visualization of five enterocyte markers. Each row represents one selected tissue section from each time point with their corresponding H&E image in the leftmost column. Spot colors represent normalized gene expression.

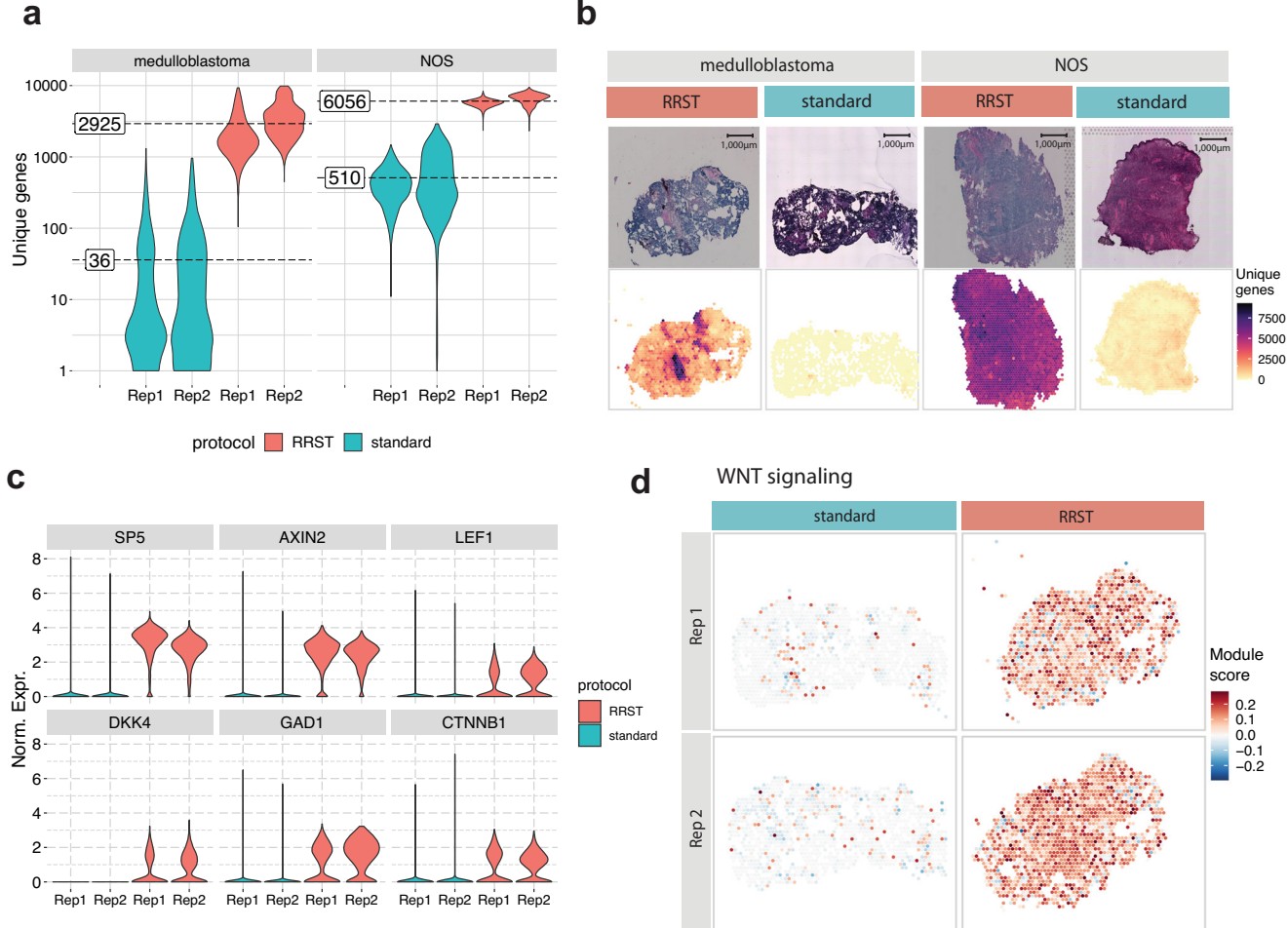

**Fig. 5 | Comparison between standard Visium and RRST protocols in eight pediatric brain tumor tissue sections. a** Violin plots showing the number of unique genes per spot in all eight tissue sections (medulloblastoma *n* = 4, NOS *n* = 4). The fill color represents the protocol used to generate the data. The average number of unique genes for each sample and protocol are highlighted by dashed lines. The *y*-axis shows log10-transformed values. **b** H&E images (top row) and the

number of unique genes per spot (bottom row) shown for four representative tissue sections. **c** Violin plots showing the normalized expression of 6 marker genes related to WNT-signaling across the four medulloblastoma tissue sections. The fill color represents the protocol used to generate the data. Rep1, replicate 1; Rep2, replicate 2. Norm. Expr., normalized gene expression. **d** Spatial visualization of WNT-signaling module scores in the WNT medulloblastoma samples.

standard Visium assay. We speculate that the underlying reason for why these experiments failed was due to either tissue detachment or inefficient permeabilization of the tissue. By applying RRST protocol to these samples, we could reach an approximate 12 to 100-fold increase in the number of detected genes per spot (Fig. 5a, b). This suggests that the RRST approach is less sensitive to changes in tissue composition compared to the standard Visium protocol.

Based on the low data quality of the standard Visium data, we were first discouraged to proceed with data analysis. However, with the RRST data, we could assess how the difference in quality affects

characterization of these tumors. For this purpose, we focused on the medulloblastoma sample, which was classified as a WNT subtype, characterized by activation of the WNT signaling pathway[21]. The medulloblastoma tissue sections were annotated by a pathologist, showing that most of the tissue sections were composed of tumor cells (Supplementary Fig. 10). To compare the data quality of RRST and standard Visium datasets obtained from the pediatric brain tumor samples, we examined the expression of WNT-signaling genes, including *AXIN2, DKK4, LEF1 and CTNNB1* and two known targets of the WNT pathway SP5, GAD1[22,23]. We were able to detect these marker

genes in the RRST dataset, but not in the standard Visium data (Fig. 5c). Moreover, we also tried estimating the WNT-signaling pathway activity by calculating a module score using a larger gene set of 42 genes[24], which detected enrichment of the pathway in the RRST data but not in the standard Visium data (Fig. 5d). These results highlight the importance of high quality data for molecular characterization of clinical samples, which for this particular sample could be achieved by RRST.

## RRST sheds light on cartilage and bone biology

Analysis of RNA profiles of cartilage and bone is a challenging task because cells in these tissues are embedded in dense extracellular matrices, which are also often mineralized[25]. Extensive enzymatic digestion is typically required to isolate cells from these tissues, but the influence of such procedure on the transcriptional profiles of these cells is not fully understood, and whether sub-populations of cells remain in the undigested tissue is typically not reported[26]. SRT offers a major advantage to study these tissues since gene expression can be analyzed without the need to isolate cells, together with the benefit of added spatial information.

The long-bones elongate via a process called endochondral ossification, in which streams of chondrocytes from the epiphyseal cartilage undergo successive differentiation stages and produce a mineralized cartilage matrix, which is subsequently remodeled and used as a scaffold on which new bone tissue is deposited[27]. One of the later developmental stages in this process is the formation of a bony structure called the secondary ossification center (SOC) within the epiphyseal cartilage[27]. In the proximal tibia of humans, this event occurs around birth[28], whereas in mice, it is precisely determined to occur between postnatal days 7 and 11[29]. Within these few days, the SOC contains many different cell-types, including osteoblasts, hematopoietic cells, mesenchymal stromal cells and endothelial cells, which are suddenly located within a few cell-diameters of the resting-zone chondrocytes, potentially influencing these cells[30]. To investigate potential effectors that derive from the newly forming SOC, we applied RRST to mouse growth plate specimens before SOC formation (postnatal day 4, P4) and immediately after SOC formation (postnatal day 11, P11)[9].

First, we aimed to benchmark our RRST protocol with the standard Visium protocol. In line with previous results on other tissue types processed in this study, we observed a 3- to 9-fold increase in the number of unique genes detected with RRST (Fig. 6a). Importantly, this trend was particularly clear in the cartilage and bone tissue, where we observed between 1298 and 1750 unique genes and between 2822 to 4000 UMIs on average with RRST, whereas the standard Visium protocol recovered less than 100 unique genes and UMIs on average (Supplementary Fig. 11a, b, e, and f). The difference in the number of genes and UMIs was also evident in the surrounding tissues, and in addition we observed more even distribution of unique genes and UMIs in the RRST datasets (Supplementary Fig. 11e, f). Based on these observations, we decided to proceed with the higher quality RRST data for downstream analysis of the cartilage and bone tissue.

Non-negative matrix factorization (NNMF) analysis identified several factors containing chondrocytes in the resting and proliferating zones (eg. *Col2a1*, *Col9a1*, Fig. 6b and Supplementary Fig. 12a)[31], hypertrophic chondrocytes and bone cells within the primary spongiosa (eg. *Col10a1*, *Mmp9*, *Phospho1*, *Dmp1*, *Acp5*, Fig. 6c and Supplementary Fig. 12b)[32,33], as well as the cruciate ligament (eg. *Scx*, *Dkk3*, Fig. 6d and Supplementary Fig. 12c)[34,35] and cells at the perichondrium/periosteum (eg. *Thbs2*, *Tnn*, Fig. 6e and Supplementary Fig. 12d)[36], which appeared in the distinct, expected anatomical locations. To explore possible secreted factors deriving from the newly forming SOC, we used the histological images to manually assign the spots within the cartilage into seven sub-clusters: "resting zone", "proliferating zone", "pre-hypertrophic", "hypertrophic zone", "SOC", "SOC-adjacent resting zone" and those surrounding the cartilage that

we grouped as "peripheral cells" (Fig. 7a and Supplementary Fig. 11d). To identify novel markers for these sub-clusters, we conducted differential gene expression analysis (Fig. 7b). We identified several genes specifically upregulated in the SOC and SOC adjacent zone; interestingly, one of these factors, *Plxnd1*, has previously been found to be expressed in newly forming ossification centers[37]. Furthermore, we identified several soluble factors that were significantly upregulated within the SOC (namely *Ccl9*, *Basp1*, and *Apln*) and SOC-adjacent zone (*Msmp*). Thus, these results show that with RRST approach we open up an exciting possibility to gain deeper understanding of bone formation and other processes occuring in the skeleton in spatial context.

## Discussion

Here we present the RNA-Rescue Spatial Transcriptomics (RRST) profiling method, designed specifically for genome-wide spatial gene expression analysis of moderate to low quality fresh frozen (FF) samples. Recent developments in the field have made it possible to generate SRT data from FFPE samples, which is the preferred fixation method for storing biological material in biobanks. Formalin-fixation provides better preservation of morphology and makes the material compatible with spatial mRNA-protein co-detection assays. While FFPE sample preservation has its advantages, overfixation leading to heavily crosslinked RNA is a common issue, which may introduce biases in the analysis of both RNA and DNA in those samples[38]. Hence, we modified the commercially available Visium FFPE spatial gene expression protocol to be applicable on FF tissues by introducing three modifications: (1) a short formalin fixation step to make RRST compatible with Visium FFPE protocol, (2) a baking step for reinforced tissue section adhesion and prevention of detachment and (3) removal of the crosslink-reversal step to prevent RNA degradation and which shortens the overall protocol time. In addition, we believe that RRST will increase flexibility for researchers working with snap-frozen samples, in particular, to make SRT compatible with other modalities that rely on FF specimens, such mass spectrometry in order to obtain paired data from the same tissue block.

In this work, we analyzed 52 tissue sections across seven different tissue types to demonstrate the versatility of RRST protocol. Although standard Visium protocol, which relies on methanol-fixation, has been shown to work in high quality FF specimens, our analysis of mouse brain and prostate cancer tissue demonstrates that RRST performs equally well in tissues with high RIN values and exhibits better performance in low-quality samples as demonstrated by the increased number of detected genes and transcripts in several different tissue types. We show that in samples collected from the human small intestine and colon, we observed severe RNA degradation in epithelial tissues; however, with the RRST protocol, we were able to recover spatial data from these tissues when the standard protocol failed.

Notably, RRST allowed us to identify characteristic WNT-signaling pathway genes in a medulloblastoma WNT subtype of pediatric brain tumors, which would have been otherwise overlooked in standard Visium-derived data. Moreover, the RRST protocol does not require tissue optimization, making it advantageous in situations where little material is available, as is often the case with precious clinical specimens. In addition, we demonstrate that RRST protocol can successfully generate transcriptomic profiles in challenging tissue types such as adult human lung or mouse cartilage/bone. For example, by applying RRST to adult human lung tissue we are able to provide a more detailed, data-driven characterization of different tissue compartments. The additional information that we observe in the RRST data makes the technology more relevant for studies of the respiratory system.

To the best of our knowledge, we have generated the first spatially resolved transcriptomics dataset from cartilage and bone tissue, which opens up new possibilities to study the composition and communication of cells in the skeletal system, for example, to better

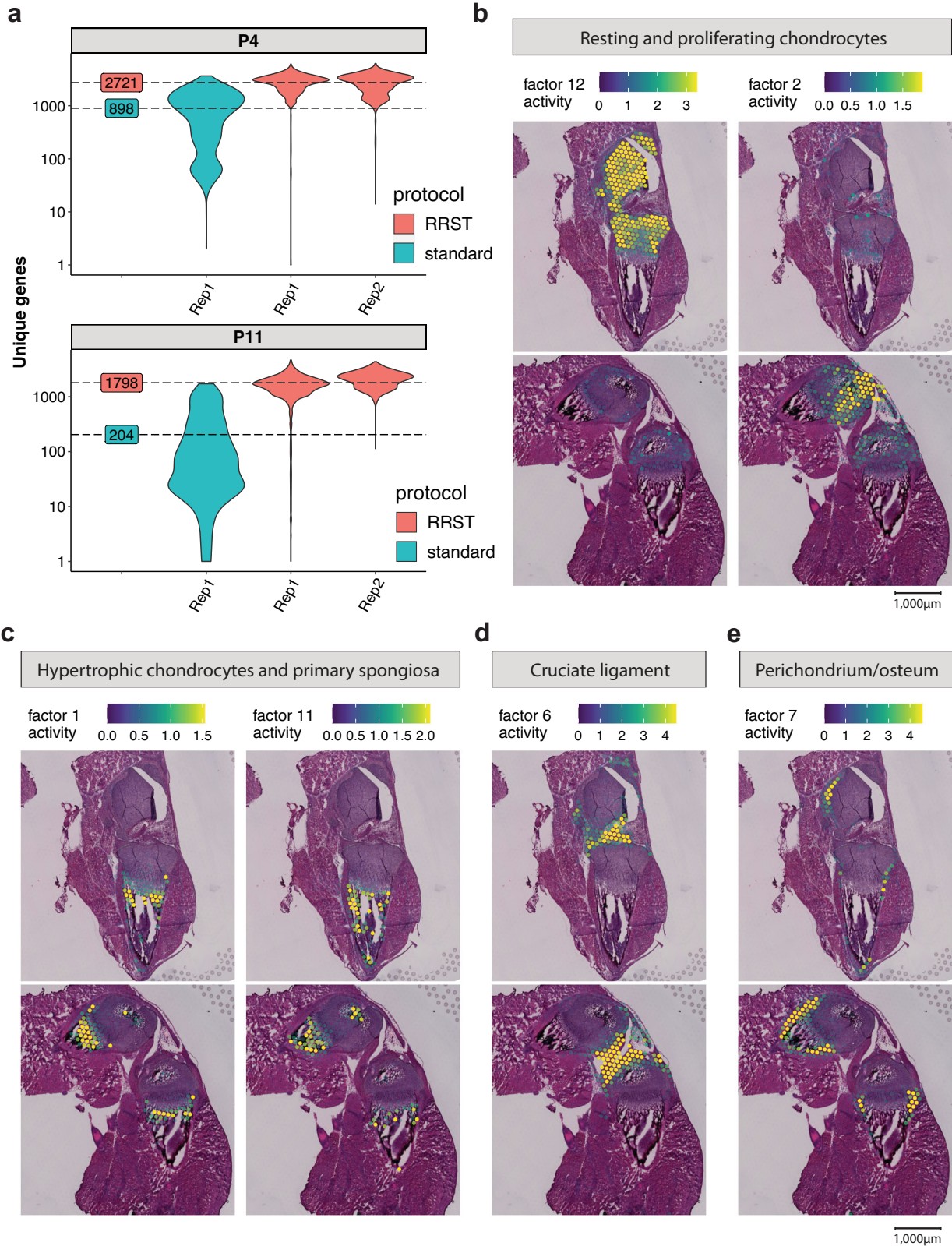

**Fig. 6 | Comparison of standard Visium with RRST on mouse cartilage tissue.**
**a** Average numbers of unique genes are highlighted by dashed lines for each protocol next to the violin plots. The *y*-axis represents log10-scaled counts. 1 tissue section was processed with the standard Visium protocol for P4 and P11, and 2 tissue sections for each timepoint (P4, P11) with the RRST protocol. **b** Following NNMF, Factors 12 and 2 associated with resting and proliferating chondrocytes.

**c** Factors 1 and 11 associated with hypertrophic chondrocytes and primary spongiosa. **d** Factor 6 associated with the cruciate ligament. **e** Factor 7 associated with Perichondrium and periosteum. Spot colors represent the factor activity, i.e., the contribution of each spot to the factor. The spot opacity has been scaled by the factor activity scores, making spots with lower scores more transparent.

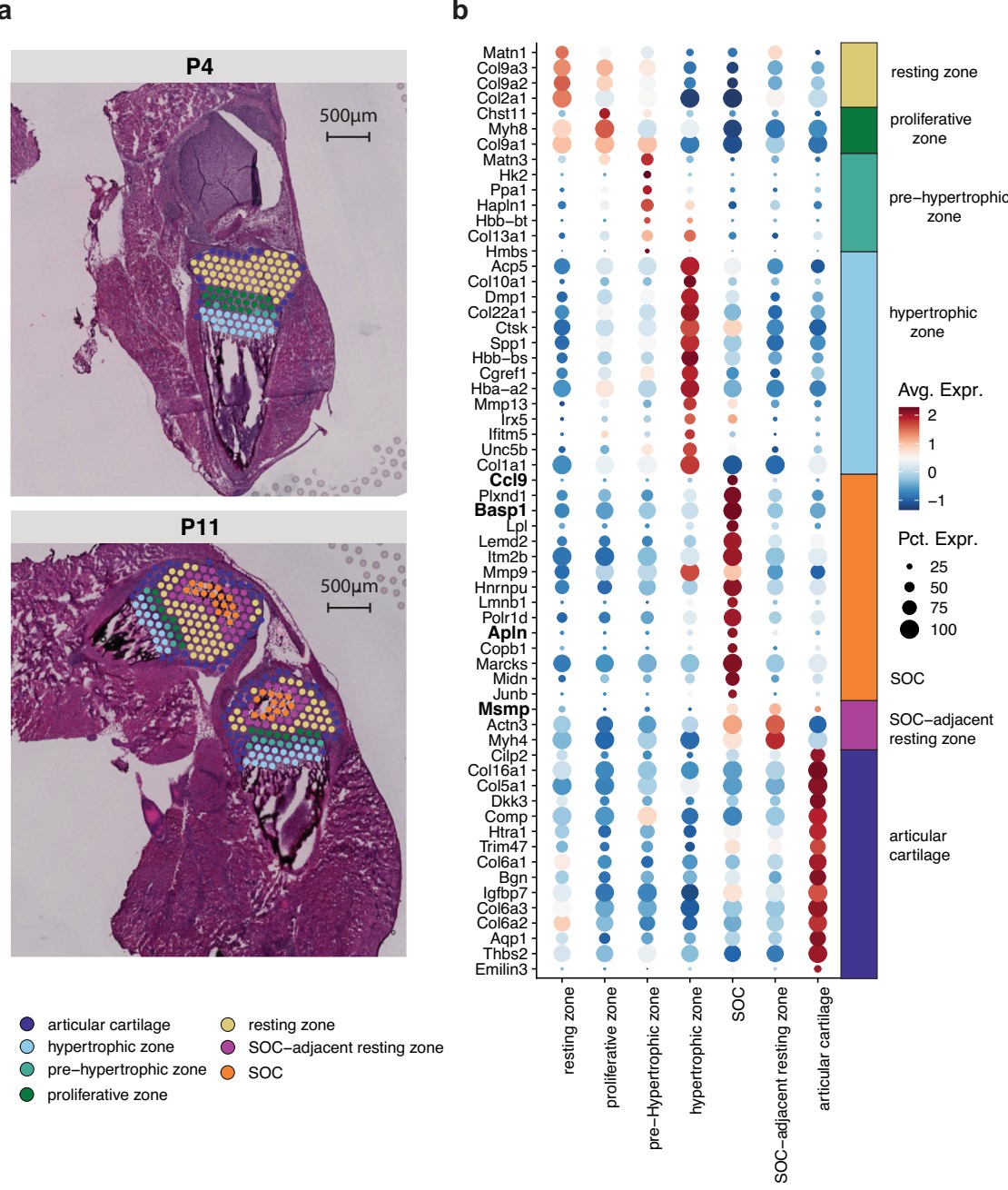

**Fig. 7 | Investigating the potential secreted markers within mouse cartilage at 2 postnatal time points. a** Comparing the manually assigned sub-clusters between postnatal day 4 and day 11. **b** Differentially upregulated genes listed from highest to lowest fold change within each annotated region (avg_log2FC > 0.6 and a maximum of 15 genes per region).

understand cellular micro-environments within the bone marrow[39], the crucial gradient of cell identities at attachment sites between muscle and bone[33], as well as to study diseases such as osteoarthritis whose step-wise progressive degeneration involves complex interplay between various tissues, including cartilage and bone[40,41]. We demonstrate that by applying RRST to mouse cartilage/bone tissue, we could identify four soluble factors expressed within the SOC or the SOC-adjacent zone, which have the potential to influence the chondrocytes, based on their close proximity. Since *Apln* has been shown to be involved in endothelial cell activation during angiogenesis[42], and *Ccl9* in the maturation of osteoclasts[43], their expression may reflect the ongoing growth and remodeling of the SOC. However, further research is required to reveal the precise roles of the four soluble factors in the SOC and their possible influence on bone growth.

In summary, we show that our RRST protocol recovers higher amounts of mRNA than the standard Visium protocol from degraded or otherwise challenging FF tissue blocks. By applying this targeted approach, we are able to obtain information about gene expression even from transcripts with fragmented or missing polyA tail, which standard Visium protocol cannot capture. Taken together, our results indicate that RRST is a powerful and versatile method, which can be used to accelerate discoveries in developmental biology, disease pathology, and clinical translational research.

## Methods

### Ethics declaration

The study was performed according to the Declaration of Helsinki, Basel Declaration, and Good Clinical Practice. All human subjects were

provided with full and adequate verbal and written information about the study before their participation. Written informed consent was obtained from all participating subjects before enrollment in the study.

Use of prostate cancer samples was approved by the Regional Ethical Review Board (REPN) Uppsala, Sweden before study initiation (Dnr 2011/066/2, Landstinget Västmanland, Sari Stenius).

Lung samples were obtained from deceased donors by the Cambridge Biorepository for Translational Medicine (CBTM) with informed consent from the donor families and approval from the NRES Committee of East of England – Cambridge South (15/EE/0152), the project has received funding from the European Union's Horizon 2020 research and innovation program under a grant agreement (no. 874656, discovAIR).

GI tract specimens were approved by the medical ethics committee of University Hospitals Leuven (approval no. S62935).

Use of pediatric brain tumor samples was approved by the Regional Ethical Review Board (EPN), Stockholm, Sweden (DNR 2018/3–31, Monica Nister).

Mouse bone samples were collected according to DNR 16673/2020, approved by Stockholm's animal experiment ethics committee (Stockholms djurförsöksetiska nämnd).

Mouse brain sample was purchased from Adlego Biomedical company, that operates under ethical permission nr. 17114–2020.

### Samples information

**Mouse brain.** A mouse brain sample was selected from a batch of commercially purchased specimens from Adlego Biomedical.

**Prostate cancer sample.** Prostate cancer sample was obtained from a surgically removed prostate at Västerås Hospital in Sweden.

**Lung specimens.** Postmortem samples from lung tissue were collected at the department of Molecular Biosciences, Science for Life Laboratory, Stockholm, Sweden. Autopsy samples were selected from two healthy donors.

**GI specimens.** Samples were collected from patients undergoing colorectal surgery. Collection of small intestine and colon samples biopsies was performed at the department of Chronic Disease and Metabolism, Katholieke Universiteit Leuven, Belgium.

**Pediatric brain tumor samples.** Samples were obtained from The Swedish Childhood Tumor Biobank.

**Mouse cartilage/bone.** Tissues were collected from postnatal mice (C57/BL6) at four and eleven days of age. Mice were group-housed with the parent mouse on a 12 h light-dark cycle at 22 °C with 50% humidity. Sex of mice was not determined due to difficulties of sex determination at these early stages of development. Briefly, hind-limbs were dissected, the skin and surrounding soft tissues were quickly trimmed. Femora and tibiae were dissected through the diaphysis and the tissue, including the knee joint, proximal tibia and distal femur (with remaining soft tissues) was embedded into OCT in a cryomold. The samples were rapidly frozen using a hexane bath.

### RNA quality evaluation

After tissue snap-freezing and prior to sample processing, 8–10 tissue sections were collected for RNA quality evaluation using the RNeasy Mini kit (Qiagen, Catalog number 74104). Extracted total RNA was measured using the Agilent Bioanalyzer (Agilent, RNA 6000 Pico kit, Part number 5067–1513) to obtain RINs. In the case of small intestine specimen, the reported RIN in this manuscript was obtained from measurements prior to the last experiments (~two years after sample collection).

### Standard Visium Spatial Gene Expression library preparation

Fresh-frozen samples were cryo-sectioned at 10 μm thickness, placed onto Visium glass slides, and stored in −80 °C before processing. Spatial gene expression libraries were generated following 10× Genomic Visium Spatial Gene Expression protocol (User Guide, CG000239 Rev F, Product number 1000187). Libraries were sequenced on Nextseq2000 (Illumina). Length of read 1 was 28 bp and read 2 150 bp.

### RRST Gene Expression library preparation

All samples used for comparison between standard Visium and RRST were collected from the same tissue blocks.

The fresh-frozen samples were cryo-sectioned at 10 μm thickness, placed onto Visium glass slides, and stored in −80 °C before processing.

1. Sample fixation and H&E staining
   - Retrieve the slide with tissue sections from −80 °C freezer and place on a thermocycler pre-heated to 37 °C for 1 min.
   - Immediately proceed to the fixation step using 4% methanol-free formaldehyde (Thermofisher, Catalog number 28906) solution for 10 min at room temperature.
   - Wash Visium slide twice in 1 × PBS (Medicago, Article number 09–9400).
   - Using a thermocycler, incubate the Visium slide at 37 °C for 20 min.
   - After incubation, wait for 5 min for the slide to cool down and proceed with tissue staining using Hematoxylin (Dako, Part number S330930-2) and Eosin (Sigma-Aldrich, Product number HT110216) (used staining times depend on tissue type).
   - Add ~100 μl of 85% Glycerol (Thermofisher, Catalog number 15514011) and, apply coverslip, proceed with tissue imaging.
   - Remove coverslip using a beaker filled with Milli-Q water.
2. Probe hybridization
   - Place Visium slide into a cassette.
   - Add 100 μl of 0.1 N HCl (Sigma-Aldrich, Product number H1758) into each well and incubate for 1 min at room temperature.
   - Remove 0.1N HCl from each well and add 100 μl of 1 × PBS to wash each well.
   - Remove 1 × PBS.
   - Immediately continue with the Pre-hybridization step according to The Visium Spatial Gene Expression for FFPE reagent kit (10× Genomics, User Guide CG000407 Rev C, mouse transcriptome Product number 1000339, human transcriptome Product number 1000338).
   - Add 100 μl of Pre-hybridization mix into each well and incubate for 15 min at room temperature.
   - At the end of incubation, remove the Pre-hybridization mix, add 100 μl of Hybridization mix.
   - Incubate Visium slide with the Hybridization mix at 50 °C overnight.
3. Probe ligation, probe release and extension, probe elution, and library preparation
   - For the rest of the library preparation, including Probe Ligation, Probe Release and Extension, Probe Elution, and FFPE Library Construction follow The Visium Spatial Gene Expression for FFPE reagent kit (10× Genomics, User Guide CG000407 Rev C, mouse transcriptome Product number 1000339, human transcriptome Product number 1000338).

Finished libraries were sequenced on Nextseq2000 (Illumina). Length of read 1 and read 2 were 28 base pairs and 50 base pairs, respectively.

### Data processing

Sequenced libraries were processed using Space Ranger software (version 1.2.1 for standard Visium data and version 1.3.1 for RRST data, 10× Genomics). Reads were aligned to the pre-built human or mouse reference genome provided by 10× Genomics (GRCh38 for human

**Table 1 | Overview of spatially resolved transcriptomics samples and filtering settings used in pre-processing steps**

| Dataset | # RRST replicates | # Standard visium replicates | # Biological replicates | Filter |
|---|---|---|---|---|
| Mouse brain | 4 | 4 | 1 | No filter |
| Human prostate cancer | 2 | 2 | 1 | No filter |
| Adult human lung | 2 | 4 | 2 | Keep spots with > 300 unique genes |
| Adult human colon | 4 | 2 | 2 | Keep spots annotated as "mucosa", "submucosa" or "muscularis" |
| Adult human small intestine | 2 | 12 | 1 | Keep spots annotated as "mucosa", "TLS", "submucosa", "muscularis" or "serosa" and spots with > 100 unique genes |
| Mouse bone | 2 | 4 | 2 | Keep spots with > 500 unique genes |
| Pediatric brain tumor | 4 | 4 | 2 | No filter |

data or mm10 for mouse data, version 32, ensembl 98), which includes a GTF file, a fasta file and a STAR index.

## Data filtering and pre-processing

Processing and analysis of spatial transcriptomics data obtained with either RRST or standard Visium was performed using R (v4.1.3) and the single-cell genomics toolkit *Seurat* and the spatial transcriptomics toolkit *STUtility*. Adult human colon and small intestine data was manually annotated into major tissue compartments based on tissue morphology (H&E image) using the interactive shiny app provided with the ManualAnnotation function in *STUtility*. Adult human colon data was categorized into three groups: "mucosa", "submucosa" and "muscularis" whereas small intestine data was categorized into five groups: "mucosa", "TLS", "submucosa", "muscularis" and "serosa". Table 1 provides a summary of the filtering settings used for each dataset. Detailed instructions for each sample type are provided in the section below.

**Mouse brain and human prostate cancer.** A total of eight mouse brain tissue sections (4xRRST and 4xstandard) and four prostate cancer tissue sections (2xRRST and 2xstandard) were used for the analysis. Spatial visualization of unique genes were created using the ST.FeaturePlot function (*STUtility*) and violin plots with the *ggplot2* R package. The median number of unique genes were calculated for each protocol and sample and visualized next to the violin plots. Gene-gene scatter plots comparing log-transformed UMI counts were created as follows: (1) raw expression matrices were extracted for each data type (RRST or standard Visium) followed by aggregating the expression values for each gene, (2) aggregated expression values were log-transformed with a pseudocount of 1 (log1p). Pearson R scores and *p*-values were calculated using the stat_cor function from the *ggpubr* R package. Gene-gene scatter plots comparing detection rates were created as follows: raw expression matrices were extracted for each data type (RRST or standard Visium) and the detection rates were estimated for each gene as the proportion of spots with detected UMI counts.

**Adult human lung.** A total of six adult human lung tissue sections (2xRRST and 4xstandard), collected from two samples were used for the analysis. After filtering out spots with fewer than 301 unique genes detected, the data was normalized and subjected to a basic analysis workflow using functions from the *Seurat* R package. The filtered data was split by sample (LNG1 and LNG1), which were analyzed separately. Normalization and scaling of the data was conducted using the NormalizeData and ScaleData functions. The top 2000 most variable genes were detected using the vst method (FindVariableFeatures) followed by dimensionality reduction by PCA (RunPCA). A shared nearest neighbor (SNN) graph was constructed based on the first 20 principal components (FindNeighbors) followed by graph-based clustering with the resolution parameter set to 0.8 (FindClusters). Finally, a Uniform Manifold Approximation and

Projection (UMAP) embedding was computed based on the first 20 principal components (RunUMAP, min.dist = 0.3, n.epochs = 1000). Marker detection was conducted by calculating differential expression for each cluster against the background (remaining clusters) with a log fold change threshold of 0.25 and an adjusted p-value threshold of 0.01 using the FindAllMarkers function. Cluster annotations were assigned based on the expression of canonical markers (obtained from a scRNA-seq atlas of the human lung[44]) and spatial co-localization with histological landmarks. Cluster marker genes shared across the two datasets (RRST and standard Visium) were selected for the following four clusters: airway epithelium, glands, smooth muscle, and megakaryocytes/platelet-enriched. Only marker genes that were identified in both datasets were considered with a maximum number of 100 markers selected per cluster. The markers were selected based on decreasing avg_log2FC values, obtained by averaging across the two conditions. Detection rates (pct. 1) were obtained from the tables produced by FindAllMarkers.

**Adult human colon.** A total of six adult human colon Visium datasets (4xRRST and 2xstandard), obtained from two samples were used for the analysis. Spots in these datasets were manually labeled using the ManualAnnotation function from *STUtility* into three major regions based on histology: "mucosa", "submucosa" and "muscularis". Unlabeled spots were removed prior to downstream analysis using the SubsetSTData function from *STUtility*. Datasets 2, 3, 4, and 6 were used for the spatial plots in Fig. 3a, b (see Table 1). Violin plots showing the distribution of unique genes in the three major regions were created for all six datasets using the *ggplot2* R package. Prior to normalization, the data was filtered to only keep genes expressed in both RRST and standard data. The dataset was then normalized using the NormalizeData function from *Seurat*. 11 intestinal epithelial marker genes were selected based on two criteria: (1) high spatial variability in the spatially resolved transcriptomics data (the data presented here), and (2) high differential expression in epithelial cells identified in the Gut Cell Atlas[18]. The normalized expression of these 11 intestinal epithelial marker genes were then visualized as violin plots for spots annotated as "mucosa".

**Human small intestine.** A total of fourteen adult human small intestine Visium datasets (2xRRST and 12xstandard), obtained from a single specimen collected over a time span of ~two years, were used for the analysis. Spots in these datasets were manually labeled using the ManualAnnotation function from *STUtility* into five major regions based on histology: "mucosa", "TLS", "submucosa", "muscularis" and "serosa". Unlabeled spots were removed prior to downstream analysis using the SubsetSTData function from *STUtility*. Violin plots showing the distribution of unique genes in the five major regions were created for all fourteen datasets using the *ggplot2* R package, with the average number of unique genes highlighted for each time point. The biotype content was calculated for ten biotypes: IG(C|J|V), TR(C|J|V), lincRNA, protein coding, mitochondrial protein coding and ribosomal protein

coding genes. All other transcripts were labeled as "other". For each biotype and within each time point, a percentage was calculated by dividing the UMIs for the biotype with the total number of UMIs. The gene annotations were obtained from the GTF file used for mapping with spaceranger. Note that the RRST protocol only targets protein coding transcripts, immunoglobulin transcripts and T-cell receptor transcripts. The average expression and detection rates were calculated for each time point for spots labeled as "mucosa". Average expression values were log10-transformed for the plot shown in Fig. 4d. Next, we split the dataset by time points, filtered out spots with less than or equal to 100 unique genes, and normalized each subset with the NormalizeData function. For DEA of the mucosa, we used the FindMarkers function to identify marker genes with a log fold change threshold of 0.25 and an adjusted $p$-value lower than 0.01 (max.cells.per.ident = 1000, ident.1 = "mucosa", only.pos = TRUE). Six intestinal epithelial marker genes were selected based on two criteria: (1) high spatial variability in the spatially resolved transcriptomics data (the data presented here), and (2) high differential expression in epithelial cells identified in the Gut Cell Atlas[18]. The normalized expression of these six intestinal epithelial marker genes were then visualized as spatial maps with ST.FeaturePlot (*STUtility*) in three selected tissue sections, one from each time point.

**Pediatric brain tumor.** A total of eight pediatric brain tumor tissue sections (4xRRST and 4xstandard), collected from two tissue blocks (medulloblastoma and NOS subtypes), were used for the analysis. The distribution of unique genes for all eight tissue sections were visualized as violin plots colored by protocol, and with the average number of unique genes highlighted next to the violin plots. One representative tissue section was selected from each combination of protocol and sample to show the distribution of unique genes together with the corresponding H&E image. Next, we downloaded cancer hallmark gene sets from MsigDB[45,46] for WNT β-catenin-signaling and TGFβ-signaling. These gene sets were then used to compute enrichment scores from the normalized medulloblastoma data with the AddModuleScore function from *Seurat*. These module scores were then visualized as spatial maps on one representative tissue section from each protocol (RRST or standard). Next, we selected six known WNT-signaling marker genes and visualized their normalized expression distributions as violin plots in the medulloblastoma data.

**Mouse bone.** A total of six tissue sections (4xRRST and 2xstandard), collected from two tissue blocks (P4 and P11), were used for the comparison shown in Fig. 6a and Supplementary Fig. 11. The spots were manually annotated into two regions: "cartilage/bone" and "surrounding" tissue. Distributions of unique genes and UMIs at the two post-natal stages and in manually annotated regions (split by protocol) were visualized with violin plots using the *ggplot2* R package and spatial maps were created with the FeatureOverlay function from *STUtility*. Only the RRST samples were used for subsequent data analysis. First, the "cartilage/bone" region was manually annotated into seven sub regions: "resting zone", "proliferative zone", "pre-Hypertrophic zone", "hypertrophic zone", "SOC", "SOC-adjacent resting zone" and "articular cartilage" (shown in Fig. 7a). Spots with at least 500 unique genes were kept prior to normalization using variance stabilizing transformation (vst) implemented in the SCTransform function from *Seurat*. The NNMF was computed on the filtered and normalized data using the RunNMF function from *STUtility*, with the number of factors set to 30. Based on visual inspection we identified eight factors colocalized with various structures of the cartilage/bone tissue region: factor_12, factor_2, factor_1, factor_11, factor_6 and factor_7 (shown in Fig. 6b−e and Supplementary Fig. 12). Next, we created a subset of the data including only the seven sub regions defined within the cartilage/bone, with the goal of extracting marker genes from each sub region by DEA. Prior to running the DEA, we first

renormalized the raw UMI counts with the NormalizeData and ScaleData functions. The DEA was conducted using FindAllMarkers from *Seurat*, while filtering out genes with adjusted $p$-values lower than 0.01 and average log fold change values higher than 0.25. Marker genes visualized in Fig. 7 were selected by keeping those with average log fold change values higher than 0.6 and maximum 15 genes per sub region.

## Statistics and reproducibility

Samples used in this study were selected based on RNA quality (RIN). No statistical method was used to predetermine sample size. In order to verify the reproducibility of the presented laboratory approach, the majority of the samples presented in this study were processed in technical replicates, here defined as consecutive sections taken from the same tissue block. Biological replicates (samples from the same tissue from two different donors) were used for adult human colon, adult human lung, mouse bone, and pediatric brain tumor (see Table 1 and Supplementary Data 1). Samples processed by standard Visium had technical or had biological replicates, with exception of a mouse bone due to the difficulty to obtain good quality data and the price for each experiment. The experiments were not randomized. All data provided was included in the data analysis. The investigators were not blinded to allocation during experiments and outcome assessment.

## Reporting summary

Further information on research design is available in the Nature Portfolio Reporting Summary linked to this article.

## Data availability

All data required to replicate the analyses, including spaceranger output files, H&E images and additional files are available at Mendeley Data with the following DOIs: "https://doi.org/10.17632/4w6krnywhn [https://data.mendeley.com/datasets/4w6krnywhn]" and "https://doi.org/10.17632/442mhsrpcm.1 [https://data.mendeley.com/datasets/442mhsrpcm/1]". Sequence data from the mouse brain and mouse bone/cartilage samples have been deposited at GEO with the accession number "GSE221571". Sequence data for the pediatric brain tumors, colon/intestine, lung and prostate samples that require controlled access following the GDPR legislation are available through a Materials Transfer Agreement with Monica Nister (monica.nister@ki.se), Guy Boeckxstaens (guy.boeckxstaens@kuleuven.be), Christos Samakovlis (Christos.Samakovlis@su.se) and Niklas Schultz (niklas.schultz@scilifelab.se), respectively. The data are available under Data Use 807 Conditions (DUO) and are limited to non-for-profit use as well as health/medical/biomedical 808 purposes. Access is granted if the above is fulfilled and local institutional review board/ethical 809 review board approvals are provided. Data coordination committees/persons will respond accordingly and timely to requests. All other relevant data supporting the key findings of this study are available within the article and its Supplementary Information files or from the corresponding author upon reasonable request. Source data are provided with this paper.

## Code availability

The code used to generate the figures, as well as instructions for running the code with a docker container, are available at https://github.com/ludvigla/RRST. A permanent version of the code is available at Zenodo: https://zenodo.org/record/7524632#.Y76J2ezMKX0[47].

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

## Acknowledgements

This project has received funding from the European Research Council (ERC) under the European Union's Horizon 2020 research and innovation

program (Discovair, grant agreement No. 782 101021019, J.L.). The study was also supported by The Swedish Cancer Society (grant agreement 71170, J.L.), Swedish Foundation for Strategic Research (grant agreement SB16-0014, J.L.), The Leona M. and Harry B. Helmsley Charitable Trust (J.L.), Swedish Childhood Cancer Fund (grant agreement PR2021-0035, J.L.) and Science for Life Laboratory (J.L.). We would like to thank Krishnaa Mahbubani for collecting human lung samples and the National Genomics Infrastructure (NGI), Sweden for providing infrastructure support. We thank Drs. Annelie Mollbrink, Alma Andersson, Eva Gracia Villacampa and Marco Vicari for helpful assistance, discussions, and reading the manuscript.

## Author contributions

R.M., Z.A., and L.L., initiated the project; R.M., Z.A., L.A.G., and X.M.A. planned and performed the experiments; L.L. analyzed all the data and generated the figures. L.K. helped analyze the pediatric brain tumor samples; P.T.N. and M.A. provided mouse bone samples, led the bone/cartilage biology part and wrote the relevant sections; A.S. undertook histopathological analysis; G.B., A.D.S., and N.S. provided adult human colon and small intestine samples; N. Schultz. provided the prostate sample. M.N. provided pediatric brain tumor samples; C.S. and A.F. provided lung samples. A.J. provided advice; R.M., Z.A., and L.L., drafted the manuscript; all authors read and approved the final manuscript. J.L. provided project guidance and supervision. L.A.G. and X.M.A. contributed equally to this work.

## Funding

## Competing interests

R.M., Z.A., L.L., L.A.G., X.A., L.K., and J.L. are scientific consultants for 10× Genomics, which holds IP rights to the ST technology. The remaining authors declare no competing interests.
