## [Peer Review File · Nature Communications]

REVIEWER COMMENTS

Reviewer #1 (Remarks to the Author):

The authors report RRST protocol, which is designed to avoid for harsh treatment from long incubation at high pH and high temperature. The harsh conditions can cause tissue detachment and low-quality data generation. The authors modified the Visium FFPE protocol to improve the spatial transcriptomics performance for fresh frozen tissues. The modifications include: (1) a short formalin fixation step, (2) a baking step for reinforced tissue section adhesion, (3) removal of the crosslink-reversal step. The authors used 52 tissue sections across 7 different tissue types to demonstrate the versatility of RRST protocol. The novelty in the method is limited, but the modifications are important for the broad research community. Many research groups can be benefited from applying this protocol, for patient samples.

Some of my comments, suggestions and questions are below:

- Figure 1c: standard Visium protocol produces 2,500 genes per spot for the brain samples. From 10x Genomics public datasets, this protocol generated more than 5000 genes per spot for fresh frozen mouse brain tissue. Why there were fewer genes reported here?
- Number of genes per spot was used as the performance matrix for comparisons between protocols. Information on the total reads per spot is also needed because these two parameters are correlated, i.e. more reads often lead to more genes detected
- Figure 2, it appears that among the two adjacent tissue sections, the one used for the standard protocol is smaller in size, and spots with lower reads/genes are in regions with either fewer cells or in the connective tissue regions (cluster 3). How would the differences in region areas and tissue types between two tissue sections affect the comparisons?
- The authors could perform statistical tests for comparisons in Fig 2b, Fig 3c, Fig 3d.
- In most cases, the authors describe the fold change in the mean/median number of genes detected. How would this be representative given the number of sample replicates and variation in the number of genes per spot?
- For the analysis of small intestine samples across time, how would the frozen storage affect the quality so dramatically just between 1 months and 6 months? Were the tissues stored at -80C and were the tissues thawed multiple times? What were the RIN values or these samples at 1 month and at 6 months storage?
- The improvement for the brain tissues was impressive. Did the tissue sections get detached? Figure 5b suggests tissue detachment after H&E imaging did not happen.
- Figure 6, the results for Rep 2 is not shown for the standard protocol. Assuming the two reps were used to run both with standard and RRST protocol. The authors should clarify this.
- In the discussion, P20 line 437: "that rely on FF specimens such as single nuclei sequencing or mass spectrometry in order to obtain paired data from the same tissue block", this is no longer true as the FFPE can also be used for single cell sequencing with the 10x CytAssist protocol.
- The authors may also discuss in more detail how the modifications led to the improvements. Avoiding detachment and reducing RNA degradation due to less harsh conditions are two reasons. What are other reasons and how these can lead to 10-100 fold improvement in gene detection?

- Minor: the use of numbers smaller than 10 and words should be consistent, for examples: seven, 5

Reviewer #2 (Remarks to the Author):

This manuscript presents an adaption of the Visium spatial transcriptomics assay that uses formaldehyde fixation to improve recovery of RNA in fresh-frozen tissue sections. This comes from a productive group that has pioneered spatial transcriptomics techniques, and the improved RNA quality and data presented here are impressive. The sample size is high: 52 tissue sections are analyzed, which is a great number for a technique that can be quite expensive. This reviewer has personally encountered some of the difficulties mentioned in the manuscript such as tissue dissociation from the slide. The degradation of RNA with time shown in Fig. 4 is also interesting.

Overall, this manuscript adds important value to the spatial transcriptomics field and most of my suggestions are minor.

Major point:

1. The manuscript states, "a step-by-step protocol can be found in the Methods section", but there is no step-by-step protocol. The Methods section should be re-written in more of a protocol format and to be more reader-friendly. For example, name the reagents used, list each step separately, and be specific about the steps that are and are not performed in which 10X Genomics protocol (i.e., give the specific user guide number and steps that are and aren't performed).

Minor points:

2. In Fig. 1, showing total UMIs/spot in addition to unique genes/spot would be informative, as is done in Fig. 2b.
3. In Figure 2b, it is initially confusing to see different scales for genes and UMIs; at first glance, UMI counts appear lower than genes.
4. In the "RNA quality evaluation", more information should be given. Was RNA extracted and scored before freezing? Afterwards? If afterwards, how long afterwards (given the degradation data shown in Fig. 4)?
5. Several graph axes need labeling; for example, the Y-axis (genes/spot) should be labeled in Fig. 5a and also Supplementary Fig. 11a,b.

Reviewer #3 (Remarks to the Author):

The authors present a modification of the standard Visium protocol for spatial transcriptomics on FF samples. The authors state that the method is "versatile, powerful, and reproducible protocol for FF specimens of different qualities and origins", which they robustly demonstrate throughout the paper with many examples and quality control. Their protocol will undoubtedly be useful for researchers in the field, particularly those studying rare clinical samples.

The manuscript does not attempt to make any biological contributions beyond a very superficial analysis of differentially expressed genes. While this is not necessary for a methods paper, the authors have data that could provide important insights (rare pediatric samples, and the first bone/cartilage dataset), and a slightly deeper biological analysis could make the paper substantially more impactful.

Introduction: Clear and complete introduction to the current state of ST methods

Results:

Line 100-115: This paragraph is not very clear, and the reasoning behind each decision is not very well explained. For example, the authors state that the long baking results in detachment from the slide in the original protocol, but also that a baking step was required for the tissue to adhere to the slide. Line 110-113 is particularly unclear. I think the paragraph could be rewritten to better explain why the modifications from the original protocol make sense in the context of FF samples.

Figure 1: The tissue architecture in 1b look slightly different for the original and the modified protocol. Would it be possible to show H&E for both in 1a? Are the sections taken from similar areas?

Line 177-181: I would not recommend comparing data quality based on the number and uniqueness of clusters obtained after dimensionality reduction, as this can be influenced by many other parameters and is difficult to interpret. The number of UMIs per spot, number of spots with >300 UMIs, and the fraction of mitochondrial transcripts, which are also described in this paragraph, are better metrics.

Line 187-190: Seems somewhat circular, as these genes were identified in the RRST data. A better test would be whether marker genes in the original protocol are even better markers in the RRST data.

Line 278-280: Similarly to the above comments, comparing sets of differentially expressed genes is problematic. To complete figure 4a and look at overlaps, I would simply characterize the overlap of genes expressed (throughout the tissue, or just in the mucosa) across conditions.

Discussion: Clear and well supported by the data. Given that the authors are experts in spatial transcriptomic methods, it would be valuable to end with their perspective on the future directions of these technologies beyond the current manuscript.

General comments:

The fact that RRST does not capture non polyA transcripts is stated, but not always phrased as a limitation. For example, line 183, non-coding transcripts are not captured, and should be phrased as a limitation rather than "Interestingly".

A more thorough examination of the biases of each method would be helpful. The fact that RRST captures only polyA transcripts is the obvious one and is addressed (though could be discussed further as mentioned above). However, there are genes (seen in figure 1d and 4d for example) which are clearly more or less captured by the two methods; could these lead to bias when analyzing data from either method?

Please keep same color map throughout the paper, including in the supplementary figures.

Reviewer #1 (Remarks to the Author):

The authors report RRST protocol, which is designed to avoid for harsh treatment from long incubation at high pH and high temperature. The harsh conditions can cause tissue detachment and low-quality data generation. The authors modified the Visium FFPE protocol to improve the spatial transcriptomics performance for fresh frozen tissues. The modifications include: (1) a short formalin fixation step, (2) a baking step for reinforced tissue section adhesion, (3) removal of the crosslink-reversal step. The authors used 52 tissue sections across 7 different tissue types to demonstrate the versatility of RRST protocol. The novelty in the method is limited, but the modifications are important for the broad research community. Many research groups can be benefited from applying this protocol, for patient samples.

Some of my comments, suggestions and questions are below:

- Figure 1c: standard Visium protocol produces 2,500 genes per spot for the brain samples. From 10x Genomics public datasets, this protocol generated more than 5000 genes per spot for fresh frozen mouse brain tissue. Why there were fewer genes reported here?

Answer: The publicly available FF mouse brain dataset from 10x Genomics represents a tissue section that was collected from a completely different tissue block. Since the dataset was selected as a showcase, it was likely collected from a very high quality sample. Furthermore, the datasets presented on the 10x Genomics website, e.g. “Mouse Brain Section (coronal)”, was sequenced at a depth of more than 100k reads per spot on average whereas our datasets were sequenced at depths ranging from 15-40k reads on average. This could also have an influence on the number of genes and UMIs detected. However, the standard Visium and RRST mouse brain datasets were sequenced at a comparable saturation of close to 95%, which is an indication that most of the cDNA libraries were quantified.

Moreover, the permeabilization time, which is not stated on the 10x Genomics website, can also influence the amount of cDNA captured. For these reasons, it is difficult to make direct comparisons between our mouse brain data and the data presented on the 10x Genomics website. We reason that the comparison between RRST and standard Visium protocol presented in our manuscript is fair since the data was generated from the same tissue block.

- Number of genes per spot was used as the performance matrix for comparisons between protocols. Information on the total reads per spot is also needed because these two parameters are correlated, i.e. more reads often lead to more genes detected

Answer: The number of reads and unique genes detected are indeed correlated but not linearly. At higher sequencing saturations, the number of unique genes only increases marginally with more sequencing reads. To make the comparison fair, we therefore attempted to match the sequencing saturation between the two protocols which can be seen in **Supplementary Table 2**. We have also clarified this in the text in the:

Performance of RRST in high quality FF samples section:

“The sequencing saturation was comparable between the two protocols (**Supplementary Table 2**).”

In order to keep the main figures as clear for the reader as possible, information about the number of read per spot is only available in **Supplementary Figure 5** and **Supplementary Table 2**: attached as a separate file

- Figure 2, it appears that among the two adjacent tissue sections, the one used for the standard protocol is smaller in size, and spots with lower reads/genes are in regions with either fewer cells or in the connective tissue regions (cluster 3). How would the differences in region areas and tissue types between two tissue sections affect the comparisons?

Answer: As the reviewer points out, the second tissue section shown in **Fig. 2a** (standard) has fewer spots because it was not centered on the capture area and is therefore cut at the bottom part. However, in this comparison we had n=1 datasets for RRST and n=2 datasets for standard Visium protocol which means that the total number of spots were higher for standard Visium. After filtering out spots with fewer than 300 unique genes, there were 2274 spots left for RRST and 1961 spots left for standard Visium. The sizes of the datasets used for downstream analyses, e.g. data-driven clustering and DEA, were therefore very similar. Since the degradation only affected specific parts of the tissue section, there is a certain bias in composition between the two datasets after filtering, where the RRST dataset contains more spots covering connective tissue and cartilage. For this reason, we focused the comparison on clusters that were easy to identify and characterize in both datasets, namely: airway epithelium, glands, smooth muscle and megakaryocyte/platelet-enriched clusters.

Although the tissues appear different in the H&E images, partly due to color differences caused by the H&E staining, they are in fact highly similar. The figure below shows a zoom view (same images as in **Fig. 2a**) in some of the areas with lowest QC metrics in the standard Visium lung dataset. The spots with lower numbers of genes are located in the cartilage (red dashed line) and surrounding connective tissues (blue dashed line), both of which are represented in the RRST and standard datasets. Since the sections were collected with some distance in between, the structures are placed in slightly different positions but the overall composition of structures is still similar.

Response Figure 1 | Zoom in view of the LNG1 tissue sections shown in Figure 2 in the manuscript. In the top row, red dashed lines represent the cartilage and blue dashed lines represent the surrounding connective tissues. In the bottom row, spots are colored red if they did not pass the QC filter threshold and light gray otherwise (see also Fig. 2 d in the manuscript).

• The authors could perform statistical tests for comparisons in Fig 2b, Fig 3c, Fig 3d.

Answer: For each of the two lung samples, we have $n=1$ datasets generated with RRST and $n=2$ datasets generated with standard Visium protocol. In order to conduct a statistical test, we would need at least $n=3$ samples. Instead, we provide a descriptive interpretation of the differences between the two data types. Generating data for consecutive tissue sections is expensive and time consuming and would likely provide

little additional value for this study. Following this line of reasoning, we prioritized demonstrating the RRST protocol on as many tissue types as possible.

- In most cases, the authors describe the fold change in the mean/median number of genes detected. How would this be representative given the number of sample replicates and variation in the number of genes per spot?

Answer: This is a valid point given the low number of replicates for certain samples and conditions. We would like to point out that we rarely observe large variation in the median number of unique genes between consecutive tissue sections processed with the sample protocol. This can be seen in **Supplementary Fig. 4** and in **Supplementary Table 2**. The fold-change estimates might be unstable, but the general trend throughout the paper should be clear: RRST consistently outperforms the standard protocol in terms of QC metrics. Accordingly, we have tried to hedge our claims about the median gene fold-changes. If the reviewers disagree with the current presentation, we will try to address this question in the text.

- For the analysis of small intestine samples across time, how would the frozen storage affect the quality so dramatically just between 1 months and 6 months? Were the tissues stored at -80C and were the tissues thawed multiple times? What were the RIN values of these samples at 1 month and at 6 months storage?

Answer: The tissue block was stored at -80°C between each experiment. For each experiment, the tissue block was exposed to temperature differences when it was placed on dry ice or sectioned in the cryostat. As mentioned in the text, we speculate that certain tissue types are more easily degraded than others during sample handling. In the case of small intestine, the measured RIN value at the experiment conducted after ~2 years from sample collection was 7.8, which is above the threshold recommended for running Visium spatial gene expression workflow (viz **Supplementary Table 1**, RIN = 7.8, DV200 = 86%). The RIN value represents an average of the entire tissue section, but we could see from the sequencing data that the degradation was spatially variable. We believe that RNA from the more stable muscle part of the tissue masked the degraded RNA in the mucosa, hence the high RIN value. RIN values at 1 and 6 months after sample collection were unfortunately not measured.

- The improvement for the brain tissues was impressive. Did the tissue sections get detached? Figure 5b suggests tissue detachment after H&E imaging did not happen.

Answer: We did not detect tissue detachment in the washing steps performed during and after the staining procedure. Although we cannot guarantee that tissue detachment did not occur in the inner parts of tissue sections without being visible to our eye. Based

on our observations, both incomplete adhesion and/or insufficient permeabilization can lead to failed data generation.

- Figure 6, the results for Rep 2 is not shown for the standard protocol. Assuming the two reps were used to run both with standard and RRST protocol. The authors should clarify this.

Answer: Based on our previous experience with bone sample processing using standard Visium protocol which resulted in low quality data and due to the price per experiment, we have processed only one tissue section per reported age (P4 and P11) using standard Visium protocol in order to have a direct comparison with RRST libraries where we generated 2 replicates per reported age to illustrate reproducibility of the reported protocol. We have clarified this in the **Figure 6** description to avoid this confusion.

- In the discussion, P20 line 437: “that rely on FF specimens such as single nuclei sequencing or mass spectrometry in order to obtain paired data from the same tissue block”, this is no longer true as the FFPE can also be used for single cell sequencing with the 10x CytAssist protocol.

Answer: This is a very good point, at the time of submitting this manuscript we were not aware of the reported protocol for single nuclei sequencing using FFPE tissues and therefore as suggested in this comment we have removed that statement from the manuscript.

- The authors may also discuss in more detail how the modifications led to the improvements. Avoiding detachment and reducing RNA degradation due to less harsh conditions are two reasons. What are other reasons and how these can lead to 10-100 fold improvement in gene detection?

Answer: We have addressed this point in the discussion section of the manuscript. We believe that in addition to the less harsh conditions and improvement in tissue attachment to the slide the targeted capture strategy used in RRST protocol, that does not rely on polyA tails of mRNAs, can lead to higher detection rate and therefore even transcripts with partially or fully degraded polyA tails can be successfully captured which is not a case in the standard Visium protocol.

- Minor: the use of numbers smaller than 10 and words should be consistent, for examples: seven, 5

Answer: We have converted some of the numbers in the manuscript text into words to be more consistent, although we left some unchanged as we believe that in some cases

writing numbers is more consistent with the standards of reporting. For example RNA Integrity Numbers (RIN) are reported as numbers even though all of them are smaller than 10. Also technical and mathematical expressions were left as numbers, for example 3' polyA, times (10 minutes), percentages and so on.

Reviewer #2 (Remarks to the Author):

This manuscript presents an adaption of the Visium spatial transcriptomics assay that uses formaldehyde fixation to improve recovery of RNA in fresh-frozen tissue sections. This comes from a productive group that has pioneered spatial transcriptomics techniques, and the improved RNA quality and data presented here are impressive. The sample size is high: 52 tissue sections are analyzed, which is a great number for a technique that can be quite expensive. This reviewer has personally encountered some of the difficulties mentioned in the manuscript such as tissue dissociation from the slide. The degradation of RNA with time shown in Fig. 4 is also interesting.

Overall, this manuscript adds important value to the spatial transcriptomics field and most of my suggestions are minor.

Major point:

1. The manuscript states, “a step-by-step protocol can be found in the Methods section”, but there is no step-by-step protocol. The Methods section should be re-written in more of a protocol format and to be more reader-friendly. For example, name the reagents used, list each step separately, and be specific about the steps that are and are not performed in which 10X Genomics protocol (i.e., give the specific user guide number and steps that are and aren't performed).

Answer: Thank you for pointing out that it would be more valuable for future users of our method to have more detailed protocol available. We have changed the “**RRST Gene Expression library preparation**” section according to your suggestion.

Minor points:

2. In Fig. 1, showing total UMIs/spot in addition to unique genes/spot would be informative, as is done in Fig. 2b.

Answer: The UMIs/spot is indeed an informative QC metric; however, these metrics are already provided in **Supplementary Fig. 5**. Since the UMIs/spot correlate with the unique genes per spots, we reasoned that it should be sufficient to present the latter. Also, we would like to avoid putting too much information into the main figures to make them as clear as possible. We hope that the reviewer agrees that it is adequate to keep the UMIs/spot distributions in the supplementary figure.

3. In Figure 2b, it is initially confusing to see different scales for genes and UMIs; at first glance, UMI counts appear lower than genes.

Answer: We have now edited **Figure 2b** so that the y-axes are shared between the unique genes and UMIs.

4. In the “RNA quality evaluation”, more information should be given. Was RNA extracted and scored before freezing? Afterwards? If afterwards, how long afterwards (given the degradation data shown in Fig. 4)?

Answer: We have addressed this point in the “**RNA quality evaluation**” section as suggested. In our standard procedure, RNA quality (RINs) are measured after snap-freezing of the tissue samples and prior to the first Visium experiment as we do not collect the samples ourselves and therefore all samples are already fresh-frozen upon arrival to our laboratory. In the case of the small intestine sample reported in this manuscript, we performed the RIN measurement before processing the samples with RRST and standard Visium protocol (~2 years after tissue collection) to obtain the most accurate RIN value in the time of sample processing with both of the protocols (RRST and standard Visium).

5. Several graph axes need labeling; for example, the Y-axis (genes/spot) should be labeled in Fig. 5a and also Supplementary Fig. 11a,b.

Answer: Labels of axis were added to the figures.

Reviewer #3 (Remarks to the Author):

The authors present a modification of the standard Visium protocol for spatial transcriptomics on FF samples. The authors state that the method is “versatile, powerful, and reproducible protocol for FF specimens of different qualities and origins”, which they robustly demonstrate throughout the paper with many examples and quality control. Their protocol will undoubtedly be useful for researchers in the field, particularly those studying rare clinical samples.

The manuscript does not attempt to make any biological contributions beyond a very superficial analysis of differentially expressed genes. While this is not necessary for a methods paper, the authors have data that could provide important insights (rare pediatric samples, and the first bone/cartilage dataset), and a slightly deeper biological analysis could make the paper substantially more impactful.

Introduction: Clear and complete introduction to the current state of ST methods

Answer: As this manuscript is fully focused on sample processing using only Visium spatial gene expression assay, we did not add other STR methods into the introduction as none of those methods are used or further discussed in the manuscript. Therefore we believe that a complete introduction to the current state of SRT methods is beyond the scope of this manuscript and by adding the extensive description of the field would expand the manuscript text beyond the required guidelines of Nature Communications. If of an interest, we referenced publications providing an overview of the ST field:

Museum of spatial transcriptomics (<https://www.nature.com/articles/s41592-022-01409-2>) and **Exploring tissue architecture using spatial transcriptomics** (<https://www.nature.com/articles/s41586-021-03634-9>). We would also like to add here a recently published paper that is not referenced in our manuscript and which provides well written overview of current SRT technologies and future perspectives called **The expanding vistas of spatial transcriptomics** (<https://www.nature.com/articles/s41587-022-01448-2>).

Results:

Line 100-115: This paragraph is not very clear, and the reasoning behind each decision is not very well explained. For example, the authors state that the long baking results in detachment from the slide in the original protocol, but also that a baking step was required for the tissue to adhere to the slide. Line 110-113 is particularly unclear. I think the paragraph could be rewritten to better explain why the modifications from the original protocol make sense in the context of FF samples.

Answer: We have re-wrote the suggested paragraph to make it more clear and less confusing for the reader. We explained that the introduced baking step is helping the

tissue adhesion to the Visium slide and therefore limiting the tissue detachment. The removal of the decrosslinking step in our RRST protocol eliminates unnecessary incubation at a high pH and temperature that could potentially harm the RNA quality in FF samples that did not undergo long formalin fixation treatment.

Figure 1: The tissue architecture in 1b look slightly different for the original and the modified protocol. Would it be possible to show H&E for both in 1a? Are the sections taken from similar areas?

Answer: Data are generated from sections collected from the same tissue blocks, but the sections are not consecutive. Instead, they were collected with some distance (~100-200 μm) apart from each other. Eight mouse brain sections (4x RRST, 4x standard Visium) and four prostate cancer sections (2x RRST, 2x standard Visium) were processed. For the purpose of simplifying **Figure 1** to make it as clear as possible, we decided to only show 1 representative H&E image for each protocol in **Figure 1**. The image below (**Response Fig. 2**) shows the four prostate cancer tissue sections stained by H&E. The H&E images can also be found in the Mendeley data repository. We hope that the reviewers agree that keeping representative H&E images is sufficient, otherwise we are willing to add H&E images for all sections shown in **Fig. 1b**.

Response Figure 2 | H&E images of the four prostate cancer tissue sections presented in Figure 1 in the manuscript. Sections 1-2 were processed with RRST and sections 3-3 were processed with standard Visium protocol. The color difference in the H&E staining is a technical artifact that is likely caused by the use of a different batch of eosin and should have no effect on the quality of the data. Although the sections are not consecutive, they were collected within a few hundred microns apart from each other and therefore represent the same major tissue structures.

Line 177-181: I would not recommend comparing data quality based on the number and uniqueness of clusters obtained after dimensionality reduction, as this can be influenced by many other parameters and is difficult to interpret. The number of UMIs per spot, number of spots with >300 UMIs, and the fraction of mitochondrial transcripts, which are also described in this paragraph, are better metrics.

Answer: Thank you for your comment, we have rephrased the text in the paragraph to explain our reasoning. As clustering and differential gene expression analysis are common steps in SRT data analysis, we applied clustering with the same settings to both RRST and standard Visium datasets to understand how the data quality can influence the data interpretation.

Line 187-190: Seems somewhat circular, as these genes were identified in the RRST data. A better test would be whether marker genes in the original protocol are even better markers in the RRST data.

Answer: We agree with the criticism that the selection of marker genes makes the comparison somewhat biased. As per the reviewers request, we have therefore changed this part of the analysis and replaced the content of **Supplementary Fig. 8** with new results. We also provide **Supplementary Fig. 8** in this document as **Response Figure 3**.

Instead of identifying genes from the RRST data, we selected the intersect of marker genes detected in both RRST and standard Visium data. Only four clusters were considered: airway epithelium, glands, smooth muscle and megakaryocyte/platelet, as these were the only clusters that were easy to identify on both conditions based on their spatial location and/or marker genes. The expression profiles of the shared marker genes are visualized as a heat map in **Supplementary Fig. 8b**. In this heat map, it is clear that the expression of shared markers is more consistent in the RRST data. As a complement to this heat map, we provide a visualization of detection rates for the marker genes within each cluster in **Supplementary Fig. 8c**. Except for the airway epithelium, the detection rates were higher for most markers in RRST data. These results highlight the difference in data quality (**Supplementary Fig. 8a**) affects the biological signal.

We have also changed the text in the “*Adult human lung tissue*” results section accordingly and we hope that these changes addresses the reviewer's concern.

Response Figure 3 (same as Supplementary Figure 8): Visualization of cluster marker genes identified in adult human lung data (LNG1). **a)** Violin plots showing the distribution of unique genes per spot within each cluster and condition. **b)** Heatmap showing the top differentially expressed genes (DEGs) selected by average log fold-changes across four annotated clusters. The color above the heat map specifies the cluster annotations and the protocol for the spots. The maximum number of selected DEGs per cluster is 100. The top 6 marker genes per cluster are highlighted next to the heat map. **c)** Gene-gene scatter plot showing detection rates for the DEGs (same as in **a)** within each cluster. Genes are colored based on whether their detection rates are higher in one of the two protocols.

Line 278-280: Similarly to the above comments, comparing sets of differentially expressed genes is problematic. To complete figure 4a and look at overlaps, I would simply characterize the overlap of genes expressed (throughout the tissue, or just in the mucosa) across conditions.

Answer: We thank the reviewer for this input and we acknowledge that comparing differentially expressed genes across conditions is indeed problematic. However, the suggestion to characterize the overlap of **genes expressed** is not straightforward since we have different numbers of sections for each time point. The number of genes expressed will largely depend on the size of the dataset, and for the dataset from ~ 6 months after sample collection we have 8 sections which will likely result in a very large number of genes detected even though the data is extremely sparse. We could circumvent this issue by sampling an even number of spots for each time point, but this would also influence the number of genes expressed.

Instead, we suggest that the relationship between the average expression and detection rates for each gene is more informative. **Response figure 3** shows this relationship at the three time points. We decided to focus on the mucosa where the difference across time points was most prominent. These plots demonstrate that both average expression and detection rates in the mucosa are lowest in the ~6 months datasets and are partially recovered by RRST in the ~ 2 years dataset. This improvement in data quality can have a substantial positive influence on downstream analysis, in particular for the detection of low abundant genes. Both metrics should be related to the number of unique genes shown in **Fig. 4b**; however, since these metrics are calculated at the gene level and not at the spot level, we argue that they provide a good complement. For example, from this plot we can assume that low abundant genes are likely to be missed in downstream analysis.

We hope that the addition of this new plot together with changes in the text are sufficient to address the comment.

Response Figure 3: relationship between average expression and detection rates in the mucosa. (Fig. 4d in the manuscript). The y axis shows log₁₀-transformed averaged expression values for each gene and the x-axis shows the gene detection rates. The detection rate is defined as the proportion of spots where the gene is expressed.

Discussion: Clear and well supported by the data. Given that the authors are experts in spatial transcriptomic methods, it would be valuable to end with their perspective on the future directions of these technologies beyond the current manuscript.

Answer: As previously mentioned in regards with the “Introduction” comment, we believe that discussing other SRT methods and their future directions are beyond the scope of this paper since the RRST protocol is specifically designed for the Visium spatial gene expression platform from 10x Genomics. For such information we would like to refer to published review papers focusing on summarizing the SRT field and addressing the challenges and future perspectives of those.

General comments:

The fact that RRST does not capture non polyA transcripts is stated, but not always phrased as a limitation. For example, line 183, non-coding transcripts are not captured, and should be phrased as a limitation rather than “Interestingly”.

Answer: We have addressed this comment in the manuscript text and stated that RRST targeted capture can be a limiting factor for certain analysis as the gene panel used in this approach only targets protein coding genes.

A more thorough examination of the biases of each method would be helpful. The fact that RRST captures only polyA transcripts is the obvious one and is addressed (through could be discussed further as mentioned above). However, there are genes (seen in

figure 1d and 4d for example) which are clearly more or less captured by the two methods; could these lead to bias when analyzing data from either method?

Answer: This is indeed an interesting venue to explore further, but we find it difficult to distinguish biases related to the different chemistries (i.e. targeted vs polyA-capture) from biases related to tissue quality. For example, if we investigate the biases in a low quality sample, we will likely find that a large fraction of genes are detected with RRST but not with standard Visium. In this case, the biases are mostly determined by the extent of RNA fragmentation which causes a dramatic reduction in capture efficiency with polyA-based capture. Thus, for most of the datasets we present in this study, we will not be able to isolate biases related to tissue quality from biases related to chemistry.

The only datasets that we think are suitable for such a comparison are the mouse brain and prostate cancer datasets that were generated from tissue blocks with high RNA integrity. For these datasets, we can assume that the differences that we see are mostly determined by the chemistry.

Unfortunately, we lack biological replicates to conduct a robust differential expression analysis to identify genes with biased expression levels. All tissue sections were collected from the same tissue blocks, both for mouse brain and for prostate cancer, and thus the number of biological replicates is $n=1$. The results from a DEA test would not be generalizable to other mouse brains or prostate cancer samples, but can only give us a rough idea about the chemistry-related biases.

The DEA was conducted as follows. First, the gene counts were aggregated for each individual tissue section and the aggregated matrix was filtered to only include genes targeted by the RRST probe panel. The DEA was conducted with the R package DESeq2. We used log fold change shrinkage with the lfcShrink method from DESeq2. At a logFoldchange threshold of 2 and adjusted p-value threshold of 0.01, we detected 1177 genes that were “up-regulated” in RRST mouse brain data and 1662 genes that were up-regulated in standard Visium mouse brain data. Many of these genes were uniquely detected in one of the two datasets (**Response Fig. 4a-b**).

These results clearly demonstrate that there are chemistry-specific biases in this dataset and that the capture efficiency of certain genes varies between the two. However, the source of these biases remains unclear. Biases can appear as a result of a number of technical factors such as: inefficient probe hybridization, off-target probe hybridization (which was recently investigated by Prakrithi et al in a preprint <https://doi.org/10.1101/2022.09.25.509336>), RNA degradation and limited diffusion of transcripts as well as biases that could arise from data processing.

Untangling the details of such biases is a highly complex task and would require substantial efforts to generate more data and conduct more analyses. With the current state of our dataset, we are not confident that the design is suitable to achieve this goal. We are also unsure if the results from the DEA presented here are interesting enough to

include in the paper, as it is difficult to generalize from the results to other datasets. We would like to avoid broadening the scope of the paper and we therefore hope that the reviewer agrees that this type of comparison is better suited for a separate study.

Response Figure 4: Differential expression analysis between RRST and standard Visium mouse brain data. a) Log fold changes vs mean of normalized counts with significant genes highlighted in red (adjusted p-value < 0.01). The top and bottom bands

indicate genes with 0 expression in one of the two data sets. **b)** Heatmap of top 25 most differentially expressed genes in RRST or standard Visium mouse brain data.

Please keep same color map throughout the paper, including in the supplementary figures.

Answer: We have now harmonized the color maps for all figures. In particular we found that there were some inconsistencies in the coloring of RRST and standard data which has now been edited in **Fig. 2** and **Fig. 5**. We have also used the same color map for all plots showing the spatial distribution of features, e.g. unique genes or number of UMIs with two exceptions. In **Fig. 5db**, the module scores are centered at 0 and we therefore chose to use a divergent color palette. In **Fig. 8b-d** and Suppl. **Fig. 11 e-f**, we did not change the color palette. Since the spatial “heatmaps” are plotted on top of H&E images, it is necessary to use a color palette that provides a good contrast against the H&E background. If the reviewer is referring to other potential inconsistencies in the figures or if we misunderstood the comment, we kindly ask for a more detailed explanation.

REVIEWERS' COMMENTS

Reviewer #1 (Remarks to the Author):

The authors' responses to my comments are sufficient. I suggest publishing this important manuscript.

Reviewer #2 (Remarks to the Author):

For my original comment #5, there is still no y-axis label in the image for Fig. 5a, although Supplementary Fig. 11a,b has been corrected.

Other than that, my concerns have been addressed.

Reviewer #3 (Remarks to the Author):

The authors have addressed all my concerns, I recommend publication of the paper.